

# Assessment of meteorology vs control measures in China fine
# particular matter trend from 2013-2019 by an environmental
# meteorology index
Sunling Gong[1], Hongli Liu[1], Bihui Zhang[2], Jianjun He[1], Hengde Zhang[2], Yaqiang Wang[1], Shuxiao
Wang[3], Lei Zhang[1], Jie Wang[4]
[1] State Key Laboratory of Severe Weather & Key Laboratory of Atmospheric Chemistry of CMA,
Chinese Academy of Meteorological Sciences, Beijing 100081, China
[2] National Meteorological Center, Beijing 100081, China
[3] School of Environment and State Key Joint Laboratory of Environment Simulation and Pollution
Control, Tsinghua University, Beijing 100084, China
[4] Zhenqi Environmental Protection Co., Lt., Hangzhou, China
**Abstract**
A framework was developed to quantitatively assess the contribution of meteorology
variations in the trend of particular matter (PM) concentrations and to separate the
impacts of meteorology from the control measures in the trend, based upon an
Environmental Meteorology Index (EMI). The model-based index EMI realistically reflects
the role of meteorology in the trend of PM and is attributed into three major factors:
deposition, vertical accumulation and horizontal transports. Based on the 2013-2019
$PM_{2.5}$ observation data and re-analysis meteorological data in China, the contributions of
meteorology and control measures in nine regions of China were assessed separately by
the EMI-based framework.  Monitoring network observations show that the $PM_{2.5}$
concentrations have been declined about 50% on national average and about 35% to 53%
for various regions. It is found that the nation-wide emission control measures were the
dominant factor in the declining trend of China $PM_{2.5}$ concentrations, contributing to
about 47% of the $PM_{2.5}$ decrease from 2013 to 2019 on the national average and 32% to



the 52% for various regions. The meteorology has a variable and sometimes critical
contribution to the year by year variations of $PM_{2.5}$ concentrations, 5% on annual average
and 10-20% for the fall-winter heavy pollution seasons.

## 1. Introduction

5       Recent observation data from the Ministry of Ecology and Environment of China (MEE)

has shown a steady improvement of air quality across the country, especially in particular
matter (PM) concentrations (Hou et al., 2019). According to 2013-2019 China Air Quality
Improvement Report issued by MEE, compared to 2013, the average concentration of
particulate matter with an aerodynamic diameter of less than 2.5 μm ($PM_{2.5}$)
concentrations in 74 major cities of China decreased by more than 50% in 2019. From
scientific and management point of views, a quantitative apportionment of the reasons
behind the trend is critical to assess the reduction strategies implemented by the
government and to guide future air quality control policy. However, the assessment of the
improvements of air quality is a complicated process that involves the quantification of
changes in the emission sources, meteorological factors, and other characteristics of the
$PM_{2.5}$ pollution, which are also interacting with each other. In order to separate the
degree of these factors, a comprehensive analysis, including observational data and
model simulation, is needed.
Researches have been done extensively on the impacts of weather systems on air
quality. Synoptic and local meteorological conditions have been recognized to influence



the PM concentrations at various scales (Beaver and Palazoglu, 2006;He et al., 2017a;He
et al., 2017b;Pearce et al., 2011a;Pearce et al., 2011b). For the atmospheric aerosol
pollution in eastern China, the dynamic effect of the downdraft in the "leeward slope" and
"weak wind area" of the Qinghai Tibet Plateau in winter is not conducive to the diffusion
of air pollution emissions in the urban agglomerations of eastern China (Xu et al., 2015;Xu
et al., 2002). The evolution of circulation situation is an important factor driving the
change of haze pollution (He et al., 2018). The local circulations, such as mountain and
valley wind and urban island circulation, have significant impact on local pollutant
concentration (Chen et al., 2009;Yu et al., 2016). Previous studies also revealed that $PM_{2.5}$
concentration is significantly correlated with local meteorological elements, such as
temperature, humidity, wind speed, and boundary layer height (He et al., 2017b;Bei et al.,
2020;Ma et al., 2019;He et al., 2016).
In the Beijing-Tianjin-Hebei (BTH) Region, a correlation analysis and principal
component regression method (Zhou et al., 2014) was used to identify the major
meteorological factors that influenced the API (Air Pollution Index) time series in China
from 2001-2010, indicating that air pressure, air temperature, precipitation and relative
humidity were closely related to air quality and resulting in a series of regression
formulas. Yet, the analysis was assumed a relatively unchanged emission whose impacts
were not taken into account. On a local scale, an attempt (Zhang et al., 2017) has been
made to correlate the air pollutant levels with a combination of meteorological factors
with the development of the Stable Weather Index (SWI) at CMA. The SWI is a composite
index which includes the advection, vertical diffusion and humidity and other



meteorological factors that are related to the formation of air pollutions in a specific
region or city. A higher value of SWI means a weaker diffusion of air pollutants. This index
had some success in assessing the meteorological impacts on air pollution, especially
calibrated for a specific region, i.e. Beijing. However, when applied to different areas
where the emission patterns and meteorological features are different, this index failed to
give a universal or comparable indication of meteorological assessment of pollution levels
across the nation.

8        Using the Kolmogorov-Zurbenko (KZ) wave filter method, Bai et al (2015) made an

effort to break the API time series in three Chinese cities into short-term, seasonal and
long-term components, and then used the stepwise regression to set up API baseline and
short-term components separately and establish linear regression models for
meteorological variables of corresponding scales. Consequently, with the long-term
representing the change of emissions removed from the time series, the meteorological
contributions alone were assumed and analyzed, pointing out that unfavorable conditions
often lead to an increase by 1-13 whereas the favorable conditions to a decrease by 2-6 in
the long-term API series, respectively. Though the contributions of emissions and
meteorological variations were separated by the research, it is only done by mathematical
transformations and far from the reality. The mechanisms behind the variation of the time
series were not investigated.

20       A chemical transport model (CTM) is an ideal tool to carry the task of assessment by

taking the meteorology, emissions and processes into considerations altogether.



Andersson et al. (2007) used a CTM to study the meteorologically induced inter-annual
variability and trends in deposition of sulphur and nitrogen as well as concentrations of
surface ozone ($O_3$), nitrogen dioxide ($NO_2$) and PM and its constituents over Europe during
1958-2001. It is found that the average European interannual variation, due to
meteorological variability, ranges from 3% for $O_3$, 5% for $NO_2$, 9% for PM, 6-9% for dry
deposition, to about 20% for wet deposition of sulphur and nitrogen. A multi-model
assessment of air quality trends with constant anthropogenic emissions was also carried
out in Europe (Colette et al., 2011) and found that the magnitude of the emission-driven
trend exceeds the natural variability for primary compounds, concluding that that
emission management strategies have had a significant impact over the past 10 years,
hence supporting further emission reductions strategies. Model assessments of air quality
trends at various regions and time periods (Wei et al., 2017;Li et al., 2015) in China were
also done and yielded some useful results. For the BTH Region, Li et al. (2015) used the
Comprehensive Air Quality Model with extensions (CAMx) plus the Particulate Source
Apportionment Technology (PSAT) and simulated the contributions of emission changes in
various sectors and changes in meteorology conditions for the $PM_{2.5}$ trend from 2006 to
2013. It was found that the change of source contribution of $PM_{2.5}$ in Beijing and northern
Hebei was dominated by the change of local emissions. However, for Tianjin, and central
and southern Hebei province, the change of meteorology condition was as important as
the change of emissions, illustrating the regional difference of impacts by meteorology
and emissions. However, the emission changes in the simulations were assumed and did
not reflect the real spatia-temporal variations.



There is no surprise that previous studies could not systematically catch the
meteorological impacts across the whole nation as the controlling meteorological factors
involving the characteristics of plenary boundary layers (PBL), wind speed and turbulence,
temperature and stability, radiation and clouds, underlying surface as well as pollutant
emissions, vary greatly from region to region. A single index or correlation cannot be
applied to the entire nation.  Obviously, in order to systematically assess the impacts of
meteorology on air pollution, these factors have to be taken into consideration in a
framework and be assessed simultaneously. This paper presents a methodology to assess
the individual impacts of meteorology and emission changes, based on a model-derived
index EMI, i.e., Environmental Meteorology Index, and observational data, providing a
comprehensive analysis of the air quality trends in various regions of China, with
mechanistic and quantitative attributions of various factors.
**2.  Methodology**
The assessment is carried out through the combination of observational data and EMI
index from model analysis. Since the emission and air quality characteristics vary greatly
from region to region in China, the analysis is divided into 9 focused regions (Figure 1).
Regional air quality dada ($PM_{2.5}$) provides the basis for the trend analysis. Separating the
trend contribution from regional emission reduction and meteorological variation needs a
framework, which is discussed below.

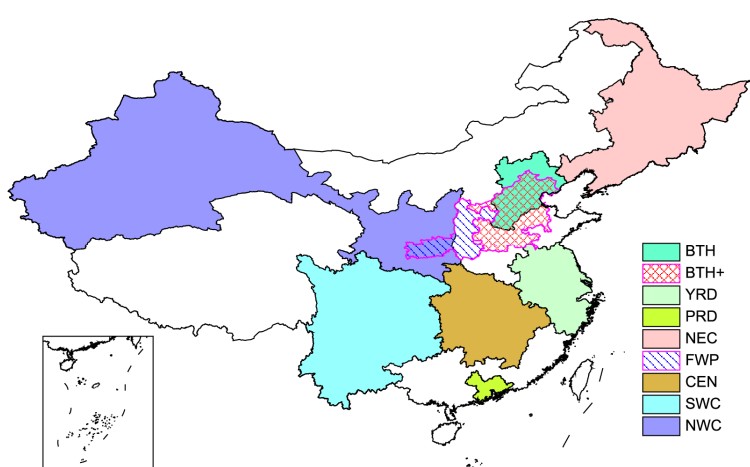

Note:**BTH**: Beijing, Tianjin and Hebei; **BTH+**: BJ, TJ + 26 cities; **YRD**: Shanghai, Jiangsu, Zhejiang and Anhui; **PRD**: 9 cities in Guangdong; **NEC**: Heilongjiang, Jilin and Liaoning; **FWP**: 11 cities in Shanxi, Shannxi and Henan; **CEN**: Hubei, Hunan and Jiangxi; **SWC**: Yunnan, Guizhou, Sichuan, Chongqing; **NWC**: Shannxi, Gansu, Ningxia and Xinjiang

Figure 1: Analysis region separation and definition.
## 2.1.        Particular Matter (PM) Observation Data
The observational pollution data of PM$_{2.5}$ concentrations used in this study were from
the monitoring network of the Ministry of Ecology and Environment (MEE) of China
(http:// english.mee.gov.cn/). From 2013 to 2019, the concentrations have shown a large
change in the country where most regions see a declined trend in the annual
concentrations. Data show that from 2013 to 2019, the national annual averaged PM$_{2.5}$
concentrations have dropped about 50% (Fig. 2), where the haze days have been
shortened by 21.2 days from the China Meteorological Administration (CMA) monitoring
data (Table 1), with some regional differences. Regionally, by 2019, the PM$_{2.5}$ reduction
rate from 2013 ranges from 35 to 53%. Detailed analysis will be given in the Results and
Discussion secession.

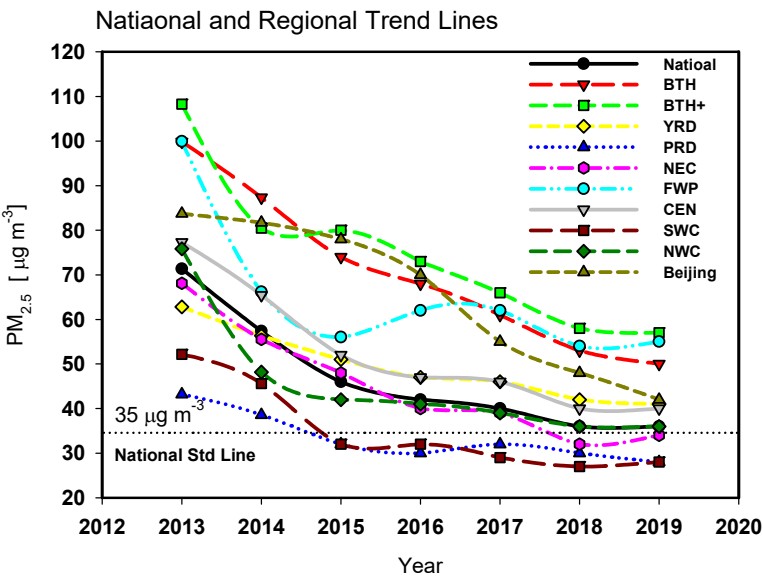

Figure 2: National and regional trend lines of $PM_{2.5}$ in China from 2013 t0 2019.

5        It is noted that the $PM_{2.5}$ mass concentrations by MEE are now reported under

observation site's actual conditions of temperature and pressure from September 1, 2018
before which the values were reported under the standard state (STP), i.e. 273 K and
101.325 kPa. In order to maintain the consistence of the data series, the $PM_{2.5}$
concentrations used in this study have all been converted according to the new standard
(MEE, 2012)(GB3095-2012) under actual conditions. Research has shown that after the
change of reporting standard, the $PM_{2.5}$ concentration in most cities decreased, and the
number of good days to meet the standard increased (Zhang and Rao, 2019).



## 2.2.    Meteorological Data

Conventional meteorological data can provide qualitative assessment of the contributions of meteorological factors to the changes of air quality. The data used in this study are from 843 national base weather stations of the CMA from 2013 to 2019. The wind speed (WS), day with small wind (DSW), relative humidity (RH) and haze days are used to analyze the pollution meteorological conditions. When the daily average wind speed is less than 2 m s$^{-1}$, a DSW day is defined. Since the haze formation is always related to stable meteorological conditions and high aerosol mass loading, haze observation from CMA is also used to analyze the haze trends and the impact of air quality on visibility. A haze day is defined with daily averaged visibility less than 10 km and relative humidity less than 85% (Wu et al., 2014), excluding days of low visibility due to precipitation, blowing snow, blowing sand, floating dust, sandstorms and smoke.

Data show that from 2013 to 2019, the national annual averaged WS has increased by 12.9%, DSW dropped by 15.1%, and RH almost unchanged (Table 1), with regional differences. It can be seen that the annual haze days have a certain degree of correlations positively with WS and negatively with DSW. Detailed analysis linking PM$_{2.5}$ and meteorology will be given in the Results and Discussion secession.



1 Table 1: National and regional environmental meteorology in 2019 and comparisons with
2 2013 and 2016

| Region | Wind Speed | | | Days with Small Wind | | | Relative Humidity | | | Haze (days) | | |
|---|---|---|---|---|---|---|---|---|---|---|---|---|
| | AVG | vs 2016 (%) | vs 2013 (%) | Days | vs 2016 (%) | vs 2013 (%) | % | vs 2016 (%) | vs 2013 (%) | Days | vs 2016 | vs 2013 |
| National | 1.9 | +3.6 | +12.9 | 129.8 | -0.8 | -15.1 | 60.1 | -6.3 | -3.9 | 25.7 | -12.1 | -21.2 |
| BTH | 1.9 | -2.7 | -2.2 | 131.0 | +5.8 | +9.0 | 56.7 | -3.3 | -4.2 | 45.2 | -15.2 | -26.1 |
| BTH+ | 2.1 | -2.2 | +0.8 | 114.4 | +4.3 | -5.6 | 58.3 | +0.2 | +0.6 | 54.5 | -21.2 | -30.3 |
| YRD | 2.1 | -2.2 | -4.7 | 114.1 | +1.9 | +5.2 | 76.3 | -1.5 | +5.5 | 34.0 | -36.9 | -54.9 |
| FWP | 2.0 | -1.6 | +10.9 | 122.8 | +7.2 | -25.2 | 59.9 | -1.5 | +3.3 | 51.6 | -32.7 | -43.8 |
| PRD | 2.0 | +0.9 | -10.4 | 118.5 | +12.7 | +14.4 | 79.7 | -3.0 | +10.3 | 3.1 | -6.4 | -34.3 |
| NEC | 2.7 | +3.6 | +12.9 | 55.8 | -10.1 | -38.4 | 61.6 | -3.4 | -5.8 | 13.6 | -19.4 | -12.4 |
| CEN | 1.8 | -2.1 | +0.4 | 172.1 | +2.2 | -2.8 | 77.9 | +0.4 | +6.9 | 30.3 | -19.7 | -23.2 |
| SWC | 1.7 | +4.4 | +12.2 | 180.7 | -6.9 | -16.3 | 74.7 | +1.5 | +5.7 | 11.1 | -13.7 | -12.4 |
| NWC | 1.9 | -3.3 | +4.3 | 146.8 | +6.3 | -9.5 | 58.5 | -0.1 | +2.8 | 20.2 | -9.6 | -6.6 |

3 **Note: "+" increased; "-" decreased**



**2.3.    EMI – the Environmental Meteorological Index**
Due to the complicated interactions of emissions, meteorology and atmospheric
processes, a single set of meteorological factors or a combination of them cannot
quantitatively attribute the individual factor to the changes of concentration
observed.
In order to quantitatively assess the impacts of meteorological conditions to the
changes of air pollution levels, an index EMI (Environmental Meteorological Index) is
defined as follows. For a defined atmospheric column (h) at a time $t$, an EMI is
defined as an indication of atmospheric pollution level:
$$\text{EMI}(t) \; = \; \text{EMI}(t0) \; + \; \int_{t0}^{t} \Delta\,\text{EMI} * \text{dt}$$

11                                                                                                 (1)

where the $\Delta$EMI is the tendency that causes the changes of pollution level in a time
interval dt defined as:
$\Delta$EMI = iEmid + i$Tran$ + i$Accu$                                                  (2)
where the iEmid is the difference between emission and deposition, and  i$Tran$ and
i$Accu$ are the net (in minus out) advection transports and the vertical accumulation
by turbulent diffusion in the column, respectively. A positive sing of each factor
indicates a net flow of pollutants into the column, and vise visa.
Mathematically, these factors are expressed as:
$$iTran = \frac{1}{C_0}\int_0^h \left( u\frac{\partial C}{\partial x} + v\frac{\partial C}{\partial y} + w\frac{\partial C}{\partial z} \right) dz$$
$iAccu = \dfrac{1}{C_0}\displaystyle\int_0^h \left[ \dfrac{\partial C}{\partial x}\left(Kx\dfrac{\partial C}{\partial x}\right) + \dfrac{\partial C}{\partial y}\left(Ky\dfrac{\partial C}{\partial y}\right) + \dfrac{\partial C}{\partial z}\left(Kz\dfrac{\partial C}{\partial z}\right) \right] dz$
$iEmid = \dfrac{1}{C_0}\displaystyle\int_0^h \left[ Emis - (V_d + L_d) \right] dz$

4                                                                          (3)

where the tendency is normalized by a factor $C_0$. For an application of EMI to the
PM$_{2.5}$, $C_0$ is set to equal 35 µg m$^{-3}$, the national standard for PM$_{2.5}$ in China(MEE,
2012), and the EMI(t) is written as EMI(t)$_{2.5}$. If the EMI$_{2.5}$ is less than 1, the
concentration level will reach or be better than the national standard.

9       It can be seen here that these key parameters account for the major

meteorological factors which control the air pollutant levels, including wind speed
and directions ($u, v, w$), turbulent mixing ($K_x, K_y, K_z$) as well as dry and wet
depositions ($V_d$ and $L_d$). Therefore, under the conditions of an unchanged emissions
(*Emis*), the *EMI* variation reflects the impacts of meteorological factors on the levels
of atmospheric pollutants. Furthermore, because of the inclusion of individual
factors such as *iTran, iAccu* and *iEmid*, the variation of *EMI(t)$_{2.5}$* can be attributed to
the variation of each factor, which gives more detailed information on the
meteorological influence to the ambient pollutant concentration variations.

18       For a period of time p (t0 to t1) when the averaged pollutant level (e.g. PM$_{2.5}$)

is compared with EMI(t)$_{2.5}$, the time integral has to be done to obtain the averaged
index for the period, such as:
$\overline{EMI(p)_{2.5}} = \dfrac{1}{t1 - t0}\displaystyle\int_{t0}^{t1} EMI(t)_{2.5}\,dt$





1          (4)

The relationship among the $\Delta$EMI, $EMI(t)_{2.5}$ and $\overline{EMI(p)_{2.5}}$ is illustrated in
Figure 3. It is clear that the $EMI(t)_{2.5}$ is a function of time and can be used to reflect
the pollution level at any time t, while the $\overline{EMI(p)_{2.5}}$ is the area under the $EMI(t)_{2.5}$
from time t0 to t1, which gives the averaged pollution levels for the period. The
derivatives of $EMI(t)_{2.5}$ are the $\Delta$EMI,  which is a positive value when the pollution is
being accumulated and a negative value when the pollution is being dispersed.

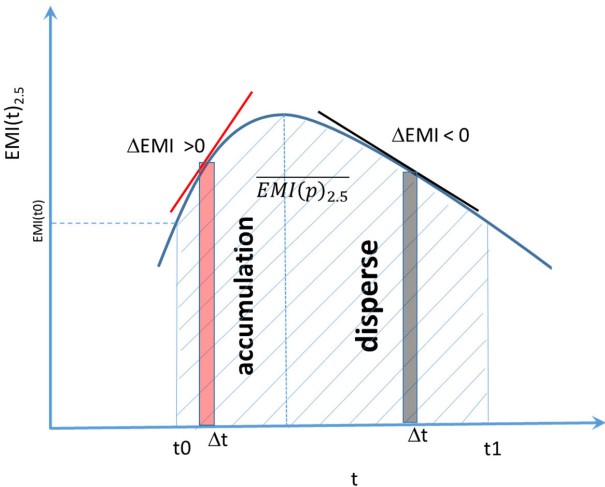

Figure 3: Relationship between the $\Delta$EMI, $EMI(t)_{2.5}$ and $\overline{EMI(p)_{2.5}}$.
Therefore, for the period p with *n* discrete steps, the $\overline{EMI(p)_{2.5}}$ represents
the averaged meteorological influences on $PM_{2.5}$, while the sum of the positive $\Delta$EMI
is the accumulation potentials and the sum of the negative $\Delta$EMI is the dispersing
potentials as illustrated in Figure 2. The relationship between them is derived as
follows:



$$\overline{EMI(p)_{2.5}} = \frac{1}{t1 - t0}[nEMI(t0)\Delta t + n\Delta EMI(1)\Delta t^2 + (n-1)\Delta EMI(2)\Delta t^2$$
$$+ (n-2)\Delta EMI(3)\Delta t^2 + (n-3)\Delta EMI(4)\Delta t^2 + \cdots + \Delta EMI(n)\Delta t^2]$$
$$\tag{5}$$
where n is the time steps in the period and the averaged EMI has been linked to the
starting point EMI(t0) and the changing rates of EMI, i.e. $\Delta EMI(n)$, at each time step.
**2.4.     Assessment Framework**
The $EMI_{2.5}$ index provides a way to assess the meteorological impacts on the
changes of $PM_{2.5}$ concentrations at two time periods, i.e. January 2103 (p0) and
January 2016 (p1) under the assumption of unchanged emissions. However, due to
the national efforts of improving air quality, the year-by-year emissions are changing
rapidly and unevenly across the country. The changes in both emissions and
meteorology tangled together to yield the observed changes in ambient
concentrations. For policy makers, the emission reduction quantification is critical to
guide the further air quality improvements. The framework proposed here is to
combined changes in the observed concentration levels and meteorology factors
$\overline{EMI(p)_{2.5}}$ to quantify the changes in emission changes at two time periods.
The observed concentrations at *p0* and *p1* are defined as PM (*m0*, *e0*) and PM
(*m1*, *e1*) where (*m0*, *e0*) and (*m1*, *e1)* indicate the meteorology and emission status
at *p0* and *p1*, respectively. The ratio of $\overline{EMI(p0)_{2.5}}/\overline{EMI(p1)_{2.5}}$ reflects the impact
ratio of sole meteorology variations on the concentrations from *p0* to *p1*. Therefore:





$$PM(m0,e1)\Big/PM(m1,e1) = \overline{EMI(p0)_{2.5}}\Big/\overline{EMI(p1)_{2.5}}$$ (6)
which gives an assumed concentration of pollutant under the conditions of
unchanged meteorology at *p0* but with new emission at *p1*. In a simple statement,
the PM (*m0*, *e1*) is the concentration at *p1* only under the influence of emission
change from *e0* to *e1* but with the same meteorology (*m0*). Consequently, the
impact of only emission changes from *e0* to *e1* on the concentration changes can be
expressed as:
$$\Delta EMIS = \frac{PM(m0,e1)-PM(m0,e0)}{PM(m0,e0)} \times 100\%$$ (7)
which can be used to assess the impact of emission changes (control measures) on
the air pollutants.

## 2.5.   Quantitative Estimate of EMI

Finally, a process-based method is developed to calculate the EMI and its
components, i.e. *iEmid*, *iTran* and *iAccu*. The main modeling frame-work used is the
chemical weather modeling system MM5/CUACE, which is a fully coupled
atmospheric model used at CMA for national haze and air quality forecasts (Gong
and Zhang, 2008;Zhou et al., 2012). CUACE is a unified atmospheric chemistry
environment with four major functional sub-systems: emissions, gas phase
chemistry, aerosol microphysics and data assimilation (Niu et al., 2008). Seven aerosol
components, i.e. sea salts, sand/dust, EC, OC, sulfates, nitrates and ammonium salts
are sectioned in 12 size bins with detailed microphysics of hygroscopic growth,
nucleation, coagulation, condensation, dry depositions and wet scavenging in the



aerosol module (Gong et al., 2003). The gas chemistry module is based on the
second generation of Regional Acid Deposition Model (RADM Ⅱ) mechanism with 63
gaseous species through 21 photo-chemical reactions and 121 gas phase reactions
applicable under a wide variety of environmental conditions especially for smog
(Stockwell et al., 1990) and prepares the sulfate and SOA production rates for the
aerosol module and for the aerosol equilibrium module ISORROPIA (Nenes et al.,
1998) to calculate the nitrate and ammonium aerosols. This is the default method to
treat the secondary aerosol formations in CUACE. For the EMI application of CUACE,
another option was also adapted to compute the secondary aerosol formations by a
highly parameterized method (Zhao et al., 2017), that computes the aerosol
formation rates directly from the pre-cursor emission rates of $SO_2$, $NO_2$ and VOC.
This option was added to facilitate timely operational forecast requirements for
CMA.  Both primary and pre-cursor emissions of PM are based on the 2016 MEIC
Inventory (http://www.meicmodel.org/) developed by Tsinghua University for China.
In order to quantitatively obtain each term defined in Equation 3, the CUACE
model was modified to extract the change rates for the processes involved. Driven
by the re-analysis meteorological data, the new system CUACE/EMI can be used to
calculate each term in ΔEMI at each time step (Δt).
In summary, this section presents a systematic platform to separate and assess
the impacts of the meteorology and emissions on the ambient concentration
changes. The $\overline{EMI(p)_{2.5}}$ and ΔEMIS form the basis for the assessment. In the Results
and Discussions section, the application of the platform is presented to assess the
fine particular matter ($PM_{2.5}$) changes in China.



**3.  Results and Discussions**
**3.1.       Validation of EMI by Observations**
Under the conditions of no changes in annual emissions for $PM_{2.5}$ and its
precursors, the daily $EMI_{2.5}$ was computed by CUACE from 2013 to 2019 on a 15×15
km resolution across China and accompanied by its contribution components: *iTran*,
*iAccu* and *iEmid*. However, in order to reflect the significant changes of industrial and
domestic energy consumptions within a year in China, a monthly emission (Wang et
al.) variation was applied to the emission inventory for computing the $EMI_{2.5}$, which
is more realistically reflecting the meteorology contributions to the $PM_{2.5}$
concentrations.
To evaluate the applicability of $EMI_{2.5}$, the index was compared with the
observed $PM_{2.5}$ concentrations. Figure 4 shows the spatial distribution of correlation
between $PM_{2.5}$ and $EMI_{2.5}$ for 2017 for all China. The correlation coefficients between
$EMI_{2.5}$ and $PM_{2.5}$ concentrations are greater than 0.4 for most of the Eastern China
and greater than 0.6 for most of the assessment regions. Less satisfactory correlation
was found in Western China, possibly due to complex terrain and less accurate
emission data over there. Furthermore, due to the uncertainty in emissions and the
difference in model performance for year-to-year meteorology simulations, the
correlation coefficients may differ for different years. Overall, the good correlation
between them merits the application of $EMI_{2.5}$ to quantify the meteorology impact
on $PM_{2.5}$.





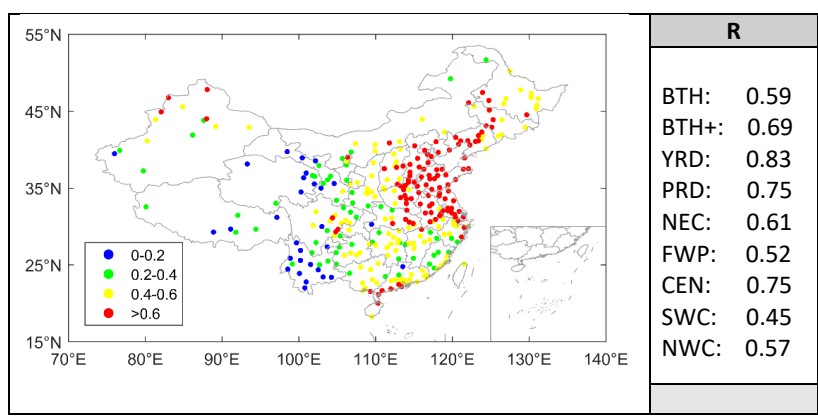

Figure 4: Correlation coefficients (R) between the $EMI_{2.5}$ and the observed $PM_{2.5}$
concentrations across China for 2017 and for typical regions averaged between 2013 and

4 2019.

6       To further illustrate the applicability of $EMI_{2.5}$, the difference of various

conditions between December 2014 and December 2015 in BTH rejoin was also
analyzed when significant change of air quality and meteorological conditions
occurred. The winter of 2015 was accompanied by a strong El Nino (ENSO) event,
resulting in significant anomalies for meteorological conditions in China. Analysis
shows that the meteorological conditions in December 2015 (compared to
December 2014) had several important anomalies, including that the surface
southeasterly winds were significantly enhanced in the North China Plain (NCP) and
the wind speeds were decreased in the middle-north of eastern China, while slightly
increased in the south of eastern China. Study suggests that the 2015 El Nino event
had significant effects on air pollution in eastern China, especially in the NCP region,
including the capital city of Beijing, in which aerosol pollution was significantly
enhanced in the already heavily polluted capital city of China (Chang et al., 2016).



1       Figure 5 shows the monthly average $EMI_{2.5}$, $PM_{2.5}$ and the contribution of

2       sub-index to total $EMI_{2.5}$ in December 2014 and 2015 over BTH region. The monthly

3       average $EMI_{2.5}$ increases about 54.9% from 2.1 in December 2014 to 3.2 in December

4       2015, indicating worsening meteorological conditions for $PM_{2.5}$ pollution. The

5       increase of $EMI_{2.5}$ is mainly contributed by adverse atmospheric transport conditions

6       (Fig. 5c), which results in the increase of $EMI_{2.5}$ reaching 3.2. With the increase of

7       background concentration, the deposition and vertical diffusion also increase, and

8       offset the impact of adverse transport conditions to some extent.

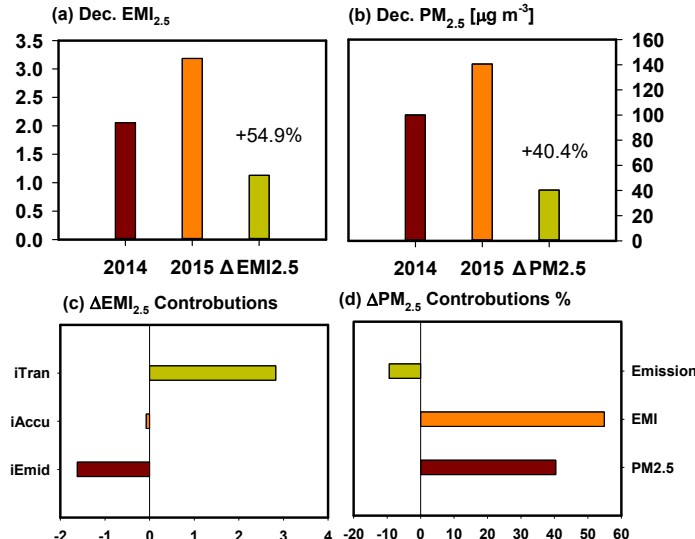

10   Figure 5: The monthly average $EMI_{2.5}$ (a), $PM_{2.5}$ (b) and the contribution of sub-index (c) and
11   contributions to the $PM_{2.5}$ changes (d) in Decembers 2014 and 2015 over BTH.

13       The worsening meteorological conditions represented by $EMI_{2.5}$ were also

14       supported by the observations for the two periods. The observed day with small

15       wind (DSW, wind speed less than 2 m s$^{-1}$) reveals that, except for part of southern





Hebei province, the DSW increases 5-15 days for 2015 in most meteorological
stations in BTH region (Fig. 6a), which indicates a large decrease of local diffusion
capability. The comparison of wind rose map shows that the decrease of northwest
wind and the increase of southwest and northeast wind occurred in December 2015
(Fig. 6b). The change of wind fields indicates more pollutants were transported to
BTH region from Shandong, Jiangsu, Henan, and Northeast China. These variations
indirectly validate the conclusions of adverse atmospheric transport conditions with
high iTran in December 2015.

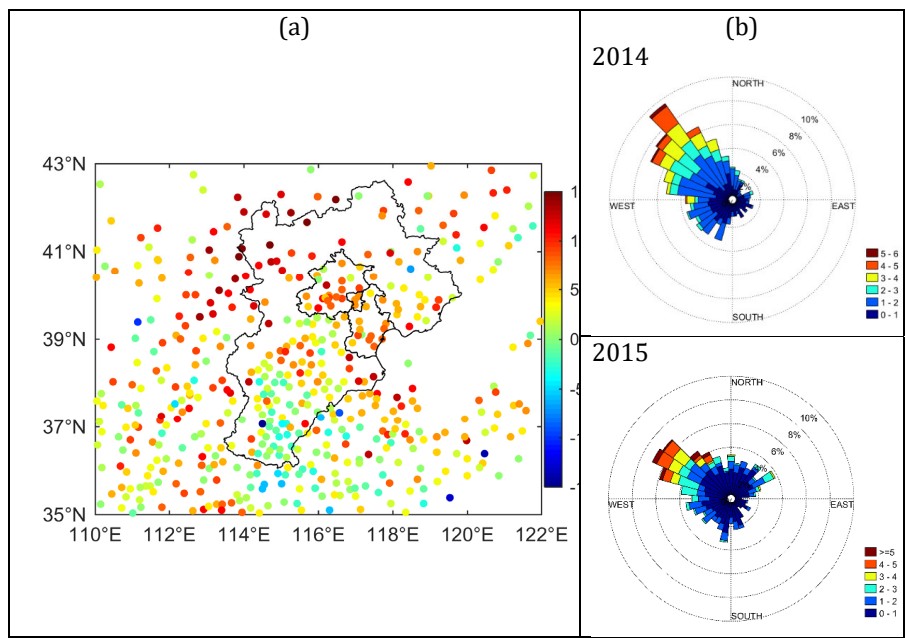

Figure 6: (a) The change of DSW (days) from December 2014 to December 2015 (December
2015 – December 2014) and (b) Wind rose maps in December 2014 and December 2015
over BTH region.

14         Based on the assessment method of emission contribution to the observed

trend (Eqs. 6 and 7), the emissions reduction in December 2015 as compared to 2014



was estimated to contribute about 9.4% (Fig. 5d) to the PM$_{2.5}$ concentration decrease,
compensating the large increase caused by meteorology, which is comparable with
previous studies of about 8.6% reduction in emissions (Liu et al., 2017;He et al., 2017a)
for the same two months. In other words, without the regional emission reduction
efforts, the observed PM$_{2.5}$ concentration in December 2015 would have had a similar
rate of 54.9% increase as the worsening meteorology conditions would bring about as
compared with December 2014. This assessment of emission reduction is supported
by the estimate of emission inventories for the BTH region in the Decembers of 2014
and 2015 by Zheng et al. (2019) who found out that the monthly emission strengths
for PM$_{2.5}$, SO$_2$, NOx, VOCs and NH$_3$ in 2015 were reduced by 22.0%, 6.9%, 2.5%, 2.5%
and 2.5%, respectively, as compared with 2014. The sensitivity and the nonlinear
response of PM$_{2.5}$ concentrations to the air pollutant emission reduction in the BTH
region (Zhao et al., 2017) have been estimated to be about 0.43 for both primary
inorganic and organic PM$_{2.5}$, 0.05 for SO$_2$, -0.07 for NOx, 0.15 for VOCs, 0.1 for NH$_3$.
Combining the emission reduction percentages between Decembers 2014 and 2015
and the nonlinear response of emissions to the PM$_{2.5}$ concentrations results in an
approximately 10.2% ambient PM$_{2.5}$ concentration reduction due to the emission
changes. This is very close to the estimate of emission reduction contribution to the
December PM$_{2.5}$ concentration difference of about 9.4% between 2014 and 2015 by
the EMI framework.


### 3.2.        PM$_{2.5}$ Trends and Meteorological Contributions
The annual averaged PM$_{2.5}$ concentrations in China have been decreased
significantly from 2013 to 2019. Figure 7 shows the observed spatial distribution of
national PM$_{2.5}$ concentrations from 2103 to 2019, respectively. These spatial
distributions are consistent with those of primary and precursor emissions of PM$_{2.5}$
(Wang et al.), pointing out to the fundamental cause of the air pollution in China.
From the spatial distributions, it is clear that the regions of BTH, FWP, CEN and NWC
had the highest PM$_{2.5}$ concentrations among the 9 regions. Even though the national
concentrations have been reduced significantly from 2013 by reducing emissions,
the pollution center of particular matters has not been changed very much, locating
at the southern Hebei Provence and indicating the macroeconomic structure has not
been gone through a great change yet. Another phenomenon can be seen from the
distribution is that in the North-west China, especially in some cities of the Xinjiang
and Ningxia Provinces, the PM$_{2.5}$ concentrations were on an increasing trend, due to
certain migrating industries from developed regions in East China.

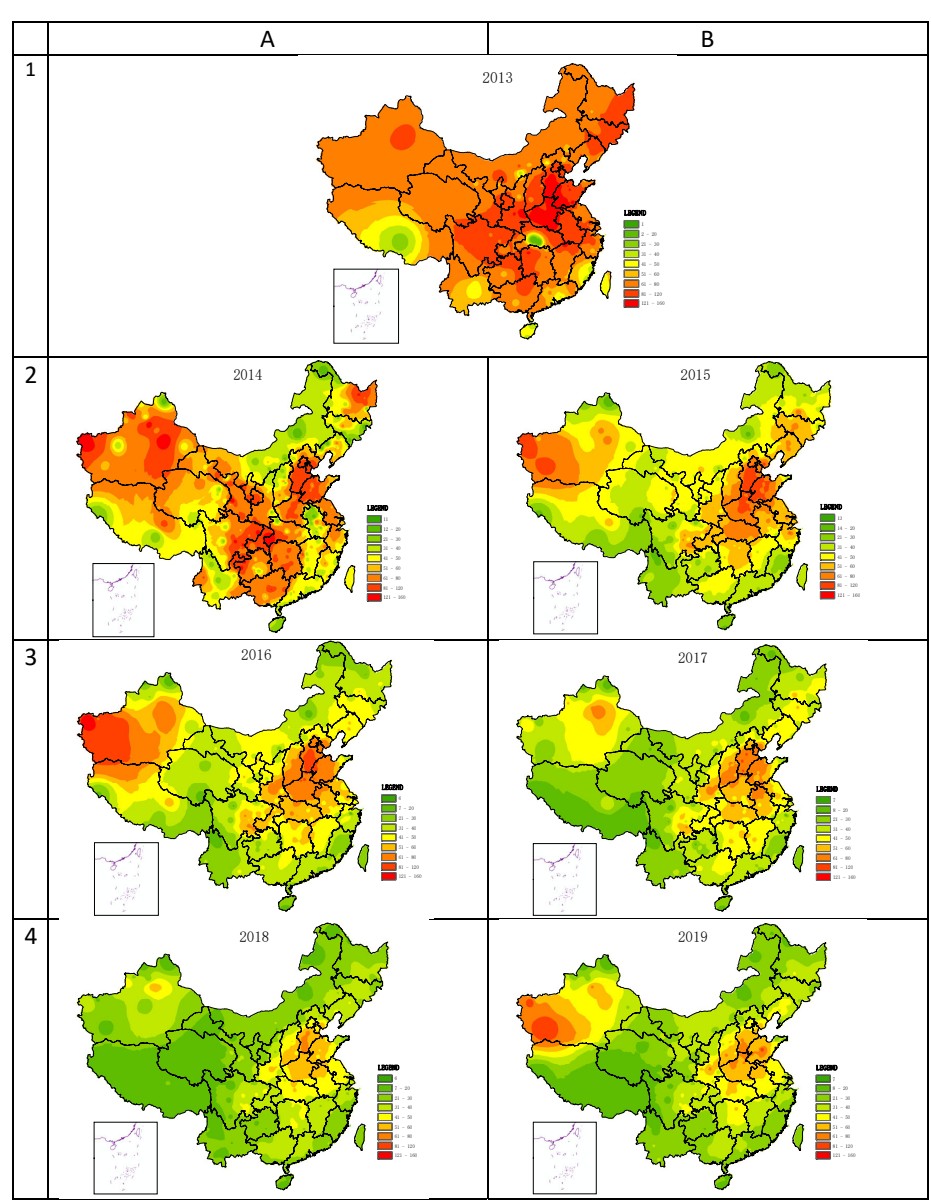

Figure 7: Regional annual PM$_{2.5}$ concentration distributions from 2013 to 2019.

4        Averaged for the nation, 9 focused regions and Beijing, the PM$_{2.5}$ trend lines

5    were shown in Figure 2. It is seen that all regions have had a large reduction of more

6    than 35% in surface PM$_{2.5}$ concentrations in 2019 as compared with those in 2013.




The averaged national annual concentration at 36 μg m$^{-3}$ has been very close to the
national standard of 35 μg m$^{-3}$ while the concentrations in PRD, SWC and NEC
regions have been below the standard. Regions above the standard are BTH+, BTH,
YRD, CEN and FWP.  Regionally, the largest drop percentage of PM$_{2.5}$ was seen in NEC
and NWC regions (Fig. 8), reaching over 50% compared with 2013. In the BTH, BTH+,
FWP and CEN regions, the reduction was in the range of 45% to 50% while in YRD
and PRD the reduction was around 35%.

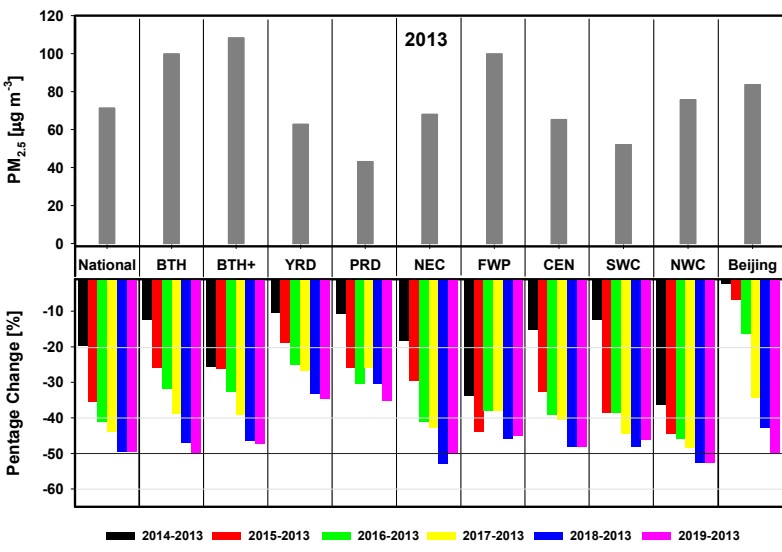

Figure 8: Annual averaged PM$_{2.5}$ concentrations in 2013 (top) and corresponding changing
rates (bottom) from 2014 to 2019 as compared with 2013 for the nation, 9 regions and
Beijing City.

13        As one of the key factors in controlling the ambient PM$_{2.5}$ concentration

variations, the annual meteorological fluctuations, i.e. EMI$_{2.5}$, from 2014 to 2019
with 2013 as the base year, are shown in Figure 9 for nine regions. Generally, the





annual $EMI_{2.5}$ shows a positive or negative variation, reflecting the meteorological
features for that specific region. Except for a couple of regions or years, most of the
fluctuations are within 5% as compared with 2013 and have a no definite trend. It
can be inferred that the meteorological conditions are possibly responsible for about
5% of the annual $PM_{2.5}$ averaged concentration fluctuations from 2013 to 2019 (Fig.
9 middle). This is consistent with what has been assessed in Europe by Andersson et
al. (2007).
The variations in meteorological contributions ($EMI_{2.5}$) to $PM_{2.5}$ for the heavy
pollution seasons of fall and winter (October 1 to March 31) generally follow the
same fluctuating pattern as the annual average but are much larger than the average
(Fig. 9 bottom), over 5% for most of the regions and years. For specific regions and
years, e.g. BTH, YRD, NEC, SWC and CEN, the variations are between 10-20% as
compared with 2013. Since the $PM_{2.5}$ concentrations are much higher in the
pollution season, the larger meteorology variations in fall-winter would exercise
more controls to the heavy pollution episodes than the annual averaged
concentrations, signifying the importance of meteorology in regulating the winter
pollution situations.
It is found that though most of the regions have a fluctuating $EMI_{2.5}$ in the
pollution season during the 2014-2019 period (Fig. 9 bottom), the YRD and FWP
show a consistent favorite and un-favorite meteorological conditions, respectively.
BTH has witnessed the same un-favorite conditions as FWP except in 2017. In other
words, in BTH and FWP, the decrease in ambient concentrations of $PM_{2.5}$ from 2014





to 2019 has to overcome the difficulty of worsening meteorological conditions with
larger control efforts.

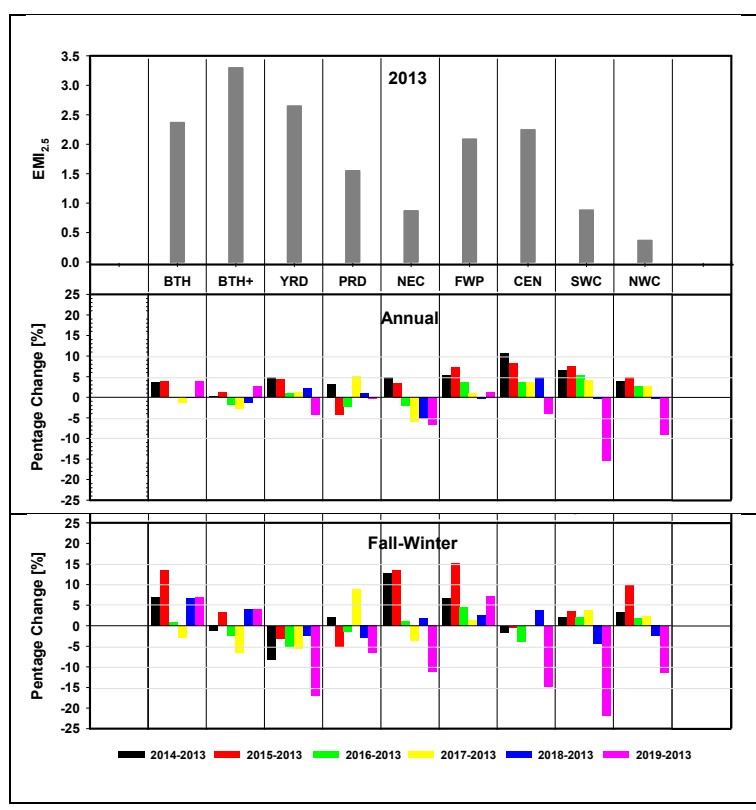

Figure 9: Annual averaged EMI$_{2.5}$ in 2013 (top) and corresponding changing rates for annual
average (middle) and for fall-winter seasons (bottom) from 2014 to 2019 as compared with
2013 in 9 regions.



### 3.3.    Attribution of Control Measures to the PM$_{2.5}$ Trend

As it is well known that the final ambient concentrations of any pollutants are resulted
from the emission, meteorology and atmospheric physical and chemical processes.
Separating emissions and meteorology contributions to the pollution level reduction entails
a combined analysis of them. The analysis in Section 3.2 shows that from 2013 to 2019, the
national averaged PM$_{2.5}$ as well as those for 9 separate regions were all showing a gradual
decline trend (Fig. 8). By 2019, 45% - 50% of reductions in surface PM$_{2.5}$ concentrations
were achieved while the meteorology contributions did not show a definite trend as from
2013, clearly pointing out the contribution of emission reductions in the trend. Using the
analysis framework for separating emissions from meteorology based on the monitoring
data of PM$_{2.5}$ and EMI$_{2.5}$ (Section 2.4), the emission change contributions are estimated.
Figure 10 shows the 2013 base emissions of PM$_{2.5}$ (Zhao et al., 2017) and the annual
changes in the emission contributions to the PM$_{2.5}$ concentrations from 2014 to 2019 as
estimated from the EMI$_{2.5}$ and observed PM$_{2.5}$. For the emissions, it is found that the unit
area emissions match better with ambient concentrations of PM$_{2.5}$ in regions than the total
emissions and the high emission regions are BTH, BTH+, YRD, PRD and FWP in 2013.
Nationally by 2019, the emission reduction contributions to the ambient PM$_{2.5}$ trend
accounted for ranging from 32% to 52% of the total PM$_{2.5}$ decrease percentage, while in BTH
and BTH+ regions the reduction was more than 49% from 2013 base year emissions, leading
the national emission reduction campaign. The emission reduction rates clearly illustrate the
effectiveness of the national-wide emission control strategies implemented since 2013 and
the emission reduction is the dominate factor for ambient PM$_{2.5}$ declining trend in China.


Taking the analysis data of PM$_{2.5}$ and EMI$_{2.5}$ from this study for BTH+ region from 2013 to
2017, it is found that control strategy contributed more than 90% to the PM$_{2.5}$ decline. Chen
et al (2019) has estimated that the control of anthropogenic emissions contributed to 80%
of the decrease in PM$_{2.5}$ concentrations in Beijing from 2013 to 2017.

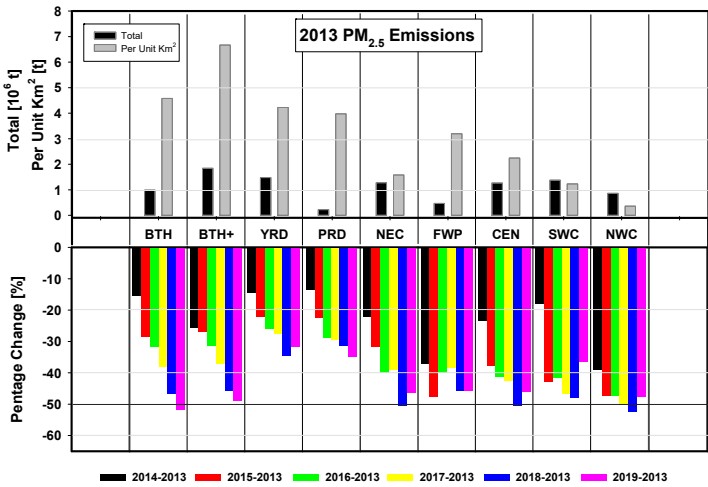

Figure 10: Annual PM$_{2.5}$ emissions (total and per unit Km$^2$) for 2013 (top) and corresponding
changing rates (bottom) from 2014 to 2019 as compared with 2013 in 9 regions.
Regionally, the emission reduction trends from 2014 to 2019 display some unique
characteristics. For the regions of BTH, BTH+ and PRD, the year-by-year reduction rate is
consistent, indicating that regardless of fluctuations in meteorology, these regions have had
an effective emission control strategy and maintained the emission reduced year by year
since 2014. However, in some regions such as FWP, NEC, SWC and NWC, the emission
reduction rates were fluctuating from 2014 to 2019, implying the emissions in these regions
were increased in certain years. Especially in FWP from 2016 to 2017, the emissions were
estimated to be increased by about 10%, and then decreased in 2018 and 2019, despite of


the factor that FWP has experienced un-favorite meteorological conditions during this
period.
Table 2 summarizes the PM$_{2.5}$ difference between 2019 and 2015 and the relative
contributions of meteorology and emission changes to the difference for all China, Beijing
and nine regions. Once again, as of the end of 2019, the PM$_{2.5}$ concentrations are all
reduced from 2015, ranging from -1.8% in FWP to -46.2% in Beijing. During this period of
time, regions of BTH, BTH+, PRD and Beijing had encountered un-favorite meteorological
conditions with positive EMI$_{2.5}$ changes, which indicated that for these regions, emission
reductions were not only to maintain the decline trend but also to offset the un- favorite
meteorological conditions in order to achieve the observed reductions in ambient PM$_{2.5}$
concentrations. On the contrary, for the regions of FWP and SWC, the emission control
impacts were to deteriorate the concentrations, implying an increase in emissions to
restrain the PM$_{2.5}$ concentration decrease by favorite meteorological conditions. For other
regions, both meteorology and emission controls contributed to PM$_{2.5}$ decrease from 2105
to 2019, with the control measures contributing from -7.9% in NWC to -68.4% in NEC (Table

16   2).

Therefore, due to the diversity of meteorological conditions and emission distributions
in China, their impacts on ambient PM$_{2.5}$ concentrations display unique reginal
characteristics. Overall, the emission controls are the dominant factor in contributing the
decline trend in China from 2013 to 2019. However, in certain regions or certain period of
years, emissions were found to increases and the meteorological dominance did occur,
which means the design of national control strategies has to take both meteorology and
emission impacts simultaneously in order to achieve maximum results.



Table 2: Observed PM$_{2.5}$ difference between 2019 and 2015 as well as its attributions to
meteorology and control measures for all China, Beijing and nine regions.

| Regions | Observed PM$_{2.5}$ Difference | | Attributions | | | |
|---|---|---|---|---|---|---|
| | | | Meteorology (EMI) | | Emission Controls | |
| | (µg m$^{-3}$) | (%) | (µg m$^{-3}$) | Relative % | (µg m$^{-3}$) | Relative % |
| National | -10 | -21.7 | -4.1 | -40.9 | -5.9 | -59.1 |
| BTH | -24 | -32.4 | +0.1 | +0.4 | -24.1 | -100.4 |
| BTH+ | -23 | -28.8 | +1.2 | +5.4 | -24.2 | -105.4 |
| YRD | -10 | -19.6 | -4.0 | -39.7 | -6.0 | -60.3 |
| PRD | -4 | -12.5 | +1.4 | +36.0 | -5.4 | -136.0 |
| NEC | -14 | -29.2 | -4.4 | -31.6 | -9.6 | -68.4 |
| FWP | -1 | -1.8 | -3.6 | -362.2 | +2.6 | +262.2 |
| CEN | -12 | -23.1 | -5.5 | -45.5 | -6.5 | -54.5 |
| SWC | -4 | -12.5 | -8.5 | -211.5 | +4.5 | +111.5 |
| NWC | -6 | -14.3 | -5.5 | -92.1 | -0.5 | -7.9 |
| Beijing | -36 | -46.2 | +3.4 | +9.4 | -39.4 | -109.4 |

**Note: "+" increased; "-" decreased**
## 4. Conclusions
Based on a 3-D chemical transport model and its process analysis, an Environmental
Meteorological Index (EMI$_{2.5}$) and an assessment framework have been developed and
applied to the analysis of the PM$_{2.5}$ trend in China from 2013 to 2019. Compared with
observations, the EMI$_{2.5}$ can realistically reflect the contribution of meteorological factors to
the PM$_{2.5}$ variations in the time series with impact mechanisms and can be used to as an
index to judge the meteorological conditions whether are favorite or not to the PM$_{2.5}$
pollutions in a region or time period. In conjunction to the observational trend data, the
EMI$_{2.5}$-based framework has been used to quantitatively assess the separate contribution of
meteorology and emission changes to the time series for 9 regions in China. Results show



1 that for the period of 2013 to 2019, the PM$_{2.5}$ concentrations have been dropped

2 continuously throughout China, by about 50% on national average. In the regions of NWC,

3 NEC, BTH, BEIJING, CEN, BTH+, SWC, the reduction was in the range of 46% to 53% while in

4 FWP, PRD and YRD, the reduction was from 45% to 35%. It is found that the control

5 measures of emission reduction are the dominant factors in the PM$_{2.5}$ declining trends in

6 various regions. By 2019, the emission reduction contributes about 47% of the PM$_{2.5}$

7 decrease from 2013 to 2019 on the national average, while in BTH region the emission

8 reduction contributes more than 50% and in YRD, PRD and SWC regions, the contributions

9 were between 32% and 37%. For most of the regions, the emission reduction trend was

10 consistent throughout the period except for FWP, NEC, SWC and NWC where the emission

11 amounts were increased for certain years. The contribution by the meteorology to the

12 surface PM$_{2.5}$ concentrations from 2013 to 2019 was not found to show a consistent trend,

13 fluctuating positively or negatively about 5% on annual average and 10-20% for the fall-

14 winter heavy pollution seasons.

16 **Acknowledgements**

17  This work was supported by the National Natural Science Foundation of China (Nos.

18 91744209, 91544232 and 41705080), and the Science and Technology Development Fund of

19 Chinese Academy of Meteorological Sciences (2019Z009).



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
