# Peer review of "Assessment of meteorology vs control measures in China fine particular matter trend from 2013-2019 by an environmental meteorology index"

_Atmospheric Chemistry and Physics, 2020_

## Referee Comment (RC1) · Kun Luo (Referee) · 2 Aug 2020

**General comments**

Based the MM5/CUACE model and observational data, an environmental meteorological index EMI2.5 and an assessment framework were developed in the present work. The roles of meteorology and control measures in China fine particular matter trend from 2013 to 2019 were separately assessed. It was found that the nationally averaged PM2.5 concentration had declined about 50

**Specific comments**

1. The current framework considers only the effects of emissions and meteorological conditions on PM 2.5 change. Actually, atmospheric chemistry plays a crucial role in shaping PM2.5 concentration. Can the authors include this factor in the framework? It would be more nice and convincing. Otherwise, the conclusions could not be so solid.

2. In the model simulations, both primary and pre-cursor emissions of PM are based on the 2016 MEIC inventory. However, the present work focused on the tendency of PM2.5 from 2013 to 2019. Did the authors use the same inventory for every year or change the inventory year by year?

3. Table 2 shows the observed PM2.5 difference between 2019 and 2015, why not 2019 and 2013 to be consistent with the title and other parts?

4. In Eq.(3) why the integration is just over dz, not dxdydz?

Technical corrections

There are numerous typos need to be corrected. I suggest the authors carefully proof read the manuscript to make sure all language problems are fixed.

1. Page 8 Line 2: "Results and Discussion secession" should be "Results and Discussion section". This should be corrected all over the entire manuscript.

2. Page 8 in Figure 2: "Natiaonal" should be "National"; "t0" should be "to"!

3. Page 11 Line 17: "sing" should be "sign"; Line 18 "vise visa" should be "vice versa"

4. Page 12 Line 11: I think Kx, Ky and Kz should be turbulent diffusion coefficients; Line 16-17: "on and to" should exchange position; Line 19 "compared" might be "computed"

5. Page 13: Line 5 and Lines 10-11. The explanation on /EMI(p)2.5 is not consistent; Line 13 "Figure 2" should be "Figure 3".

6. Page 14: Lines 7 to 9, why January 2103 (should be 2013!) to January 2016?; Line

15 "combined" should be "combine".

7. Page 17: Lines 3-10, I could not understand what the authors meant. Line 8 (Wang et al.) is not a proper citation; Line 13 the introduction of Figure 4 is not consistent with the actual caption in Page 18 Lines 2-4.

8. Page 19: In Figure 5 the subcaptions of (c) and (d), "contributions" should be "contributions".

9. Page 22: Line 4 "2103" should be "2013"; Line 6 (Wang et al.) is not a proper citation; "to" should be removed from "pointing out to".

10. Page 25: Line 20, "favorite and un-favorite" might be "favorable and unfavorable". This is also true for other statements hereafter.

11. Page 29: Line 14 "2105" should be "2015"!

12. Page 30: Line 10, to judge whether the meteorological conditions are favorite or not to...

---

## Referee Comment (RC2) · Anonymous Referee #2 · 4 Aug 2020

Referee Report for "Assessment of meteorology vs control measures in China fine particular matter trend from 2013-2019 by an environmental meteorology index" by Sunling Gong et al. (manuscript #: acp-2020-348)

It has been a long-standing problem to quantify the roles of meteorology and emission change in regional air pollution variations. Different modeling tools and techniques have been developed and utilized to address this problem. In this study, the authors developed a process analysis-based framework in a chemical transport model named CUACE to identify the driving factors of $PM_{2.5}$ changes in China during 2013-2019. They defined an Environmental Meteorological Index (EMI) by tracking the contributions of different physical processes including transport, diffusion, emission, and deposition to simulated $PM_{2.5}$ concentrations in the model. The topic is within the scope of the journal and the research question is of broad interest in the community. In general, the manuscript is well-structured, but the English writing in some parts (especially the methodology section) can be improved for clear description. Based on the current version, I have some major concerns about the theoretical basis of this EMI framework. Please see below the detailed comments to be addressed.

(1) The EMI-based analytical framework is based on the continuity equation and is similar to some mature and widely-used probing tools in other CTMs such as the integrated process rate (IPR) analysis in CMAQ and the process analysis (PA) tool in CAMx. However, the major difference between this EMI method and other probing tools is that secondary aerosol formation is missing in the EMI framework. Let's revisit a simplified continuity equation (Seinfeld and Pandis, 2016) and compare it with EMI defined in this study:

$$\frac{\partial \langle c \rangle}{\partial t} + \bar{u}_j \frac{\partial \langle c \rangle}{\partial x_j} = \frac{\partial}{\partial x_j}\left(K_{jj}\frac{\partial \langle c \rangle}{\partial x_j}\right) + R + S$$

Above is the atmospheric diffusion equation that is based on the mixing-length theory and two assumptions (negligible molecular diffusion and incompressible atmosphere). CTMs, including CUACE used in this study, use this equation to describe the spatiotemporal evolution of air pollutant concentrations. According to the EMI definition in Section 2.3 of this study, EMI is a time integral of atmospheric pollution changing tendency that consists of three parts: *iTran*, *iAccu*, and *iEmid* (Eqs. (1)-(3) on page 11/12). The first part *iTran* corresponds to the advection term on the LHS of the diffusion equation but with an opposite sign (after moving the advection term from the LHS to the RHS), the second part *iAccu* corresponds to the turbulent diffusion term (much greater than molecular diffusion) on the RHS of the diffusion equation, and the third part *iEmid* corresponds to the last source and sink term *S* on the RHS of the diffusion equation. Since ΔEMI only includes these three parts without the chemical generation term *R* (which also depends on meteorological factors such as temperature and relative humidity), it only approximates the direct physical processes modulating aerosol concentrations and ignores other meteorological impacts on chemical reactions and secondary formation of $PM_{2.5}$.

Given the large contribution of secondary formation to PM$_{2.5}$ concentrations in China (Huang et al., 2014), it's inevitable to include the comprehensive aerosol processes regarding chemical formation, nucleation, condensation, coagulation, and gas–particle partitioning etc. in this kind of probing tools to conserve the mass balance in CTMs. For instance, the IPR analysis in CMAQ considers the effects of individual physical processes and the net effect of chemistry (aerosol processes) on gas-phase air pollutants (PM$_{2.5}$). It also provides more details of the chemical transformations associated with the model's chemical mechanism in the integrated reaction rate (IRR) analysis. It's noted that the CUACE model considers secondary aerosol formation in its aerosol module (line 7-8 on page 16), but it uses "a highly parameterized method" to directly estimate secondary aerosol formation from precursors including SO$_2$, NO$_2$, and VOC for the EMI application (line 8-11 on page 16). Therefore, it seems the EMI framework can only take account of the three listed physical processes (*iTran*, *iAccu*, and *iEmid*) without consideration of meteorological effects on secondary inorganic and organic aerosol formation and transformation, which is a limitation of the current framework that should be pointed out in the manuscript.

(2) Even focusing on EMI itself without considering complex aerosol processes, the EMI framework is still problematic to be applied for assessing meteorological contributions to PM$_{2.5}$ concentration changes. A simple way to demonstrate this is to consider two idealized extreme conditions: the first is an extreme stagnation case with zero wind and the second is an extreme dispersion case with single-direction high winds (time invariant). In both cases we assume no precipitation, no wet deposition ($L_d = 0$ in *iEmid*), and negligible dry deposition compared with emissions ($Emis \gg V_d$ in *iEmid*).

In the first stagnation case, the first two parts of $\Delta EMI$ (*iTran* and *iAccu*) would diminish to zero since there is no wind (no advection) and no turbulence (no turbulent diffusion). The third part *iEmid* would be dominated by the constant emission term (assumed in line 12 on page 12) given $Emis \gg V_d$ and $L_d = 0$. In this case, $\Delta EMI$ approximates to an emission-based constant that is irrelevant to meteorology ($\Delta EMI = iTran + iAccu + iEmid = \frac{1}{c_0} \int_0^h (Emis) dz = \frac{1}{c_0} Emis \cdot h$, where $c_0$, $Emis$, and $h$ are constants). After applying this approximation to Eqs. (5)-(6) in the manuscript, the ratio of $\overline{EMI(p0)_{2.5}} / \overline{EMI(p1)_{2.5}} = EMI(p0)_{2.5} / \left( EMI(p0)_{2.5} + \frac{1}{c_0} Emis \cdot h \cdot (p1 - p0) \right)$ becomes a variable that only depends on the initial value of EMI ($EMI(p0)_{2.5}$), scaling constant $c_0$, constant $Emis$ intensity, vertical height $h$, and the time interval between $p0$ and $p1$, which does not reflect the meteorological impact on PM$_{2.5}$ concentration changes from $p0$ and $p1$ (though in this case the meteorological impact should be zero as all considered meteorological processes have been turned off or neglected and PM$_{2.5}$ concentrations solely depends on emission intensity and time intervals) as alleged in line 19-20 on page 14.

In the second case with extreme dispersion conditions, $\Delta EMI$ would be dominated by the first advection term *iTran* due to constant high winds (the concentration gradient still exists because of the constant emission source), and $\overline{EMI(p1)_{2.5}}$ would keep increasing to a huge number after a long time integral of $\Delta EMI$. Given a predetermined initial value

of $\overline{EMI(p0)_{2.5}}$ at $p0$, the ratio of $\overline{EMI(p0)_{2.5}}/\overline{EMI(p1)_{2.5}}$ in Eq. (6) approaches to zero after a long time because of the much greater denominator, which again fails to represent the meteorological impact on concentration changes from $p0$ and $p1$ (in this case the right answer for the meteorological impact should be ~100% because of the dominant role of strong advection, while the emission impact reduces to nearly zero).

The failure of the EMI framework to describe meteorological impacts on PM$_{2.5}$ concentrations results from the incorrect inclusion of emissions ($Emis$) in EMI, which contradicts its objective to separate meteorological effects from emissions. Given such defect in its theoretical basis, there is no need to further discuss the EMI-based modeling results.

Below are some technical corrections and comments:

(1) What are the units of EMI and $\Delta EMI$? Is EMI unitless as shown in Fig. 5/9? You will get different answers after doing dimensional analysis for Eqs. (1)-(3).
(2) How to determine the initial value for EMI(t0)? Here I assume t0 denotes the first day of 2013, which is the start point of the model simulation. But the initial value for EMI is not mentioned in the manuscript.
(3) What is $h$ in Eq. (3)? Is it boundary layer height or not?
(4) What kind of data were used for the correlation in Fig. 4? Monthly? Or Daily?
(5) The time intervals for model evaluation are inconsistent throughout the manuscript. For example, Figs. 7-10 show the comparison from 2013 to 2019, but Table 2 shows the attribution results between 2015 and 2019.

Reference

Seinfeld, J. H., and Pandis, S. N.: Atmospheric chemistry and physics: from air pollution to climate change, John Wiley & Sons, Inc, Hoboken, New Jersey, 2016.

Huang, R., Zhang, Y., Bozzetti, C. et al. High secondary aerosol contribution to particulate pollution during haze events in China. Nature 514, 218–222, https://doi.org/10.1038/nature13774, 2014.

---

## Author Comment (AC1) · 17 Sep 2020

General comments Based the MM5/CUACE model and observational data, an environmental meteorological index EMI2.5 and an assessment framework were developed in the present work. The roles of meteorology and control measures in China fine particular matter trend from 2013 to 2019 were separately assessed. It was found that the nationally aver- aged PM2.5 concentration had declined about 50

Specific comments 1. The current framework considers only the effects of emissions

and meteorological conditions on PM 2.5 change. Actually, atmospheric chemistry plays a crucial role in shaping PM2.5 concentration. Can the authors include this factor in the framework? It would be more nice and convincing. Otherwise, the conclusions could not be so solid.

Reply: The same question was raised by another referee. The answer to the question is as follows:

The EMI index was focused explicitly on three major physical processes of iTran, iAccu, and iEmid that are closely related to the meteorological influences. However, the secondary formation of aerosols is implicitly considered in the EMI as the three major physical processes are calculated from the concentrations of aerosols (C) as indicated in Equation (3), which are resulted from the full processes of chemical mechanisms or "a highly parameterized method" that accounts for the secondary aerosol formations. Furthermore, we have done a comparison of simulated PM2.5 with full processes and the EMI with the parameterized method, and the correlation coefficients between them range 0.72 to 0.93 for the regions in this study. The limitation of non-inclusion of explicit chemical terms in the EMI is pointed out in the manuscript.

2. In the model simulations, both primary and pre-cursor emissions of PM are based on the 2016 MEIC inventory. However, the present work focused on the tendency of PM2.5 from 2013 to 2019. Did the authors use the same inventory for every year or change the inventory year by year?

Reply: In order to isolate the meteorological impacts, we have used the 2016 MEIC emissions throughout the simulations, i.e. from 2013 to 2019, resulting the differences caused by meteorological changes only.

3. Table 2 shows the observed PM2.5 difference between 2019 and 2015, why not 2019 and 2013 to be consistent with the title and other parts?

Reply: The same question was raised by another referee. The answer to the question

is as follows:

There are two issues here that prompted us to use two different time intervals for the comparisons in the paper. The first issue is the completeness of the network observational data series of PM2.5 in China. The systematical and network observations of PM2.5 started in China from 2013. However, it took about two years (until 2015) to develop to the current status. Number of monitoring stations national-wide in 2013 was less than 900, reached to about 1400 in early 2015 and maintained the same up to now. To show the completeness of the observed PM2.5 time series and for most part of the paper, we made the comparison starting from 2013 as graph illustrations. The second issue is the data consistence and policy relevance of the assessment. Statistically, because the national observation site numbers are relative constant from 2015 to 2019, it makes more sense to use the 2015-2019 data for numerical assessment such as those shown in Table 2. The use of 2015-2019 data for Table 2 was also motivated by the introduction of the Environmental Protection Law of People's Republic of China in January 2015. For the regulation assessment point of view, the comparison Table 2 was in line with the date of the law introduction and the impact assessment by emission changes was more relevant to the interests of management to show how effective the law was.

4. In Eq.(3) why the integration is just over dz, not dxdydz?

Reply: The Equation (3) was introduced to account for the column loading of aerosols in the PBL, that contains most of the aerosol masses, for a grid. The dxdy, i.e. advection terms, is done in the 3-D transport part of the model.

Technical corrections There are numerous typos need to be corrected. I suggest the authors carefully proof read the manuscript to make sure all language problems are fixed.

1. Page 8 Line 2: "Results and Discussion secession" should be "Results and Discussion section". This should be corrected all over the entire manuscript.

Reply: Thanks. Corrected!

2. Page 8 in Figure 2: "Natiaonal" should be "National"; "t0" should be "to"!

Reply: Thanks. Corrected!

3. Page 11 Line 17: "sing" should be "sign"; Line 18 "vise visa" should be "vice versa"
Reply: Thanks. Corrected!

4. Page 12 Line 11: I think Kx, Ky and Kz should be turbulent diffusion coefficients; Line 16-17: "on and to" should exchange position; Line 19 "compared" might be "computed"

Reply: Done for the "on and to" exchange. Line 19 "compared" was not changed as we indeed meant to compare.

5. Page 13: Line 5 and Lines 10-11. The explanation on /EMI(p)2.5 is not consistent; Line 13 "Figure 2" should be "Figure 3".

Reply: Thanks. Corrected!

6. Page 14: Lines 7 to 9, why January 2103 (should be 2013!) to January 2016?; Line 15 "combined" should be "combine".

Reply: Thanks. Corrected!

7. Page 17: Lines 3-10, I could not understand what the authors meant. Line 8 (Wang et al.) is not a proper citation; Line 13 the introduction of Figure 4 is not consistent with the actual caption in Page 18 Lines 2-4.

Reply: As in the Question 2 above, we have used the MEIC emissions for 2016 for all simulations, but applied a monthly variation based on Wang et al. 2011: (Verification of anthropogenic emissions of China by satellite and ground observations). We have checked with the authors of this paper and been assured that the monthly variations were discussed.

We have corrected the Line 13 to "spatial distribution of correlation coefficients

between..."

8. Page 19: In Figure 5 the subcaptions of (c) and (d), "contributions" should be "contributions".

Reply: Thanks. Corrected! The caption for Figure 5 has been re-written.

9. Page 22: Line 4 "2103" should be "2013"; Line 6 (Wang et al.) is not a proper citation; "to" should be removed from "pointing out to".

Reply: Thanks. Corrected!

10. Page 25: Line 20, "favorite and un-favorite" might be "favorable and unfavorable". This is also true for other statements hereafter.

Reply: Thanks. Corrected: 4 in total.

11. Page 29: Line 14 "2105" should be "2015"!

Reply: Thanks. Corrected!

12. Page 30: Line 10, to judge whether the meteorological conditions are favorite or not to. . .

Reply: Thanks. Corrected!

---

## Author Comment (AC2) · 17 Sep 2020

Replies to Questions by Referee 2

It has been a long-standing problem to quantify the roles of meteorology and emission change in regional air pollution variations. Different modeling tools and techniques have been developed and utilized to address this problem. In this study, the authors developed a process analysis-based framework in a chemical transport model named CUACE to identify the driving factors of PM2.5 changes in China during 2013-2019.

They defined an Environmental Meteorological Index (EMI) by tracking the contributions of different physical processes including transport, diffusion, emission, and deposition to simulated PM2.5 concentrations in the model. The topic is within the scope of the journal and the research question is of broad interest in the community. In general, the manuscript is well-structured, but the English writing in some parts (especially the methodology section) can be improved for clear description. Based on the current version, I have some major concerns about the theoretical basis of this EMI framework. Please see below the detailed comments to be addressed.

Reply: The methodology section has been revised to clarify some descriptions, which may have caused certain confusions for the referee to raise the questions (1) and (2) below.

[Due to equations and special characters used in the questions and replies, please see the full reply in the Supplement by PDF]

Please also note the supplement to this comment:
https://acp.copernicus.org/preprints/acp-2020-348/acp-2020-348-AC2-supplement.pdf

[Figure]

**Fig. 1.** Illustration of relationship between 3 levels of EMI definition for the cases suggested by Referee #2

**Supplement:**

Replies to Questions by Referee 2

It has been a long-standing problem to quantify the roles of meteorology and emission change in regional air pollution variations. Different modeling tools and techniques have been developed and utilized to address this problem. In this study, the authors developed a process analysis-based framework in a chemical transport model named CUACE to identify the driving factors of PM2.5 changes in China during 2013-2019. They defined an Environmental Meteorological Index (EMI) by tracking the contributions of different physical processes including transport, diffusion, emission, and deposition to simulated PM2.5 concentrations in the model. The topic is within the scope of the journal and the research question is of broad interest in the community. In general, the manuscript is well-structured, but the English writing in some parts (especially the methodology section) can be improved for clear description. Based on the current version, I have some major concerns about the theoretical basis of this EMI framework. Please see below the detailed comments to be addressed.

Reply: The methodology section has been revised to clarify some descriptions, which may have caused certain confusions for the referee to raise the questions (1) and (2) below.

(1)     The EMI-based analytical framework is based on the continuity equation and is similar to some mature and widely-used probing tools in other CTMs such as the integrated process rate (IPR) analysis in CMAQ and the process analysis (PA) tool in CAMx. However, the major difference between this EMI method and other probing tools is that secondary aerosol formation is missing in the EMI framework. Let's revisit a simplified continuity equation (Seinfeld and Pandis, 2016) and compare it with EMI defined in this study:

$$\frac{\partial \langle c \rangle}{\partial t} + \bar{u}_j \frac{\partial \langle c \rangle}{\partial x_j} = \frac{\partial}{\partial x_j}\left(K_{jj}\frac{\partial \langle c \rangle}{\partial x_j}\right) + R + S$$

Above is the atmospheric diffusion equation that is based on the mixing-length theory and two assumptions (negligible molecular diffusion and incompressible atmosphere). CTMs, including CUACE used in this study, use this equation to describe the spatiotemporal evolution of air pollutant concentrations. According to the EMI definition in Section 2.3 of this study, EMI is a time integral of atmospheric pollution changing tendency that consists of three parts: iTran, iAccu, and iEmid (Eqs. (1)-(3) on page 11/12). The first part iTran corresponds to the advection term on the LHS of the diffusion equation but with an opposite sign (after moving the advection term from the LHS to the RHS), the second part iAccu corresponds to the turbulent diffusion term (much greater than molecular diffusion) on the RHS of the diffusion equation, and the third part iEmid corresponds to the last source and sink term S on the RHS of the diffusion equation. Since ΔEMI only includes these three parts without the chemical generation term R

(which also depends on meteorological factors such as temperature and relative humidity), it only approximates the direct physical processes modulating aerosol concentrations and ignores other meteorological impacts on chemical reactions and secondary formation of PM2.5. Given the large contribution of secondary formation to PM2.5 concentrations in China (Huang et al., 2014), it's inevitable to include the comprehensive aerosol processes regarding chemical formation, nucleation, condensation, coagulation, and gas–particle partitioning etc. in this kind of probing tools to conserve the mass balance in CTMs. For instance, the IPR analysis in CMAQ considers the effects of individual physical processes and the net effect of chemistry (aerosol processes) on gas-phase air pollutants (PM2.5). It also provides more details of the chemical transformations associated with the model's chemical mechanism in the integrated reaction rate (IRR) analysis. It's noted that the CUACE model considers secondary aerosol formation in its aerosol module (line 7-8 on page 16), but it uses "a highly parameterized method" to directly estimate secondary aerosol formation from precursors including SO2, NO2, and VOC for the EMI application (line 8-11 on page 16). Therefore, it seems the EMI framework can only take account of the three listed physical processes (iTran, iAccu, and iEmid) without consideration of meteorological effects on secondary inorganic and organic aerosol formation and transformation, which is a limitation of the current framework that should be pointed out in the manuscript.

Reply: Thank you for pointing this issue out. The same question was raised by another referee. The answer to the question is as follows:

The EMI index was focused explicitly on three major physical processes of iTran, iAccu, and iEmid that are closely related to the meteorological influences. However, the secondary formation of aerosols is implicitly considered in the EMI as the three major physical processes are calculated from the concentrations of aerosols (C) as indicated in Equation (3), which are resulted from the full processes of chemical mechanisms or "a highly parameterized method" that accounts for the secondary aerosol formations. Furthermore, we have done a comparison of simulated PM2.5 with full processes and the EMI with the parameterized method, and the correlation coefficients between them range 0.72 to 0.93 for the regions in this study, indicating that the parameterized method used in this study for EMI largely approximates the variation of PM2.5 with full processes. The limitation of non-inclusion of explicit chemical terms in the EMI is pointed out in the manuscript.

(2)      Even focusing on EMI itself without considering complex aerosol processes, the EMI framework is still problematic to be applied for assessing meteorological contributions to PM2.5 concentration changes. A simple way to demonstrate this is to consider two idealized extreme conditions: the first is an extreme stagnation case with zero wind and the second is an extreme dispersion case with single-direction high winds (time invariant). In both cases we

assume no precipitation, no wet deposition ($L_d$ = 0 in iEmid), and negligible dry deposition compared with emissions ($Emis \gg V_d$ in iEmid).

In the first stagnation case, the first two parts of $\Delta EMI$ (iTran and iAccu) would diminish to zero since there is no wind (no advection) and no turbulence (no turbulent diffusion). The third part iEmid would be dominated by the constant emission term (assumed in line 12 on page 12) given $Emis \gg V_d$ and $L_d$ = 0. In this case, $\Delta EMI$ approximates to an emission-based constant that is irrelevant to meteorology ($\Delta EMI = iTran + iAccu + iEmid = 1/C_0 \int (Emis)dz = 1/C_0 Emis \cdot h$, where $C_0$, $Emis$, and $h$ are constants). After applying this approximation to Eqs. (5)-(6) in the manuscript, the ratio of

$$\overline{EMI(p0)_{2.5}}/\overline{EMI(p1)_{2.5}}=EMI(p0)_{2.5}/\left(EMI(p0)_{2.5} +\frac{1}{c_0}Emis\cdot h\cdot(p1-p0)\right)$$

becomes a variable that only depends on the initial value of EMI ($EMI(p0)_{2.5}$), scaling constant $C_0$, constant $Emis$ intensity, vertical height $h$, and the time interval between $p0$ and $p1$, which does not reflect the meteorological impact on PM$_{2.5}$ concentration changes from $p0$ and $p1$ (though in this case the meteorological impact should be zero as all considered meteorological processes have been turned off or neglected and PM2.5 concentrations solely depends on emission intensity and time intervals) as alleged in line 19-20 on page 14.

In the second case with extreme dispersion conditions, $\Delta EMI$ would be dominated by the first advection term iTran due to constant high winds (the concentration gradient still exists because of the constant emission source), and $\overline{EMI(p1)_{2.5}}$ would keep increasing to a huge number after a long time integral of $\Delta EMI$. Given a predetermined initial value of $\overline{EMI(p0)_{2.5}}$ at $p0$, the ratio of $\overline{EMI(p0)_{2.5}}/\overline{EMI(p1)_{2.5}}$ in Eq. (6) approaches to zero after a long time because of the much greater denominator, which again fails to represent the meteorological impact on concentration changes from $p0$ and $p1$ (in this case the right answer for the meteorological impact should be ~100% because of the dominant role of strong advection, while the emission impact reduces to nearly zero).

The failure of the EMI framework to describe meteorological impacts on PM$_{2.5}$ concentrations results from the incorrect inclusion of emissions ($Emis$) in EMI, which contradicts its objective to separate meteorological effects from emissions. Given such defect in its theoretical basis, there is no need to further discuss the EMI-based modeling results.

Reply: To answer the questions, we first have to clarify three levels of EMI definitions:

Level 1: $\Delta$EMI, the tendency that causes the changes of pollution level at each time step $\Delta t$.

Level 2: EMI($t$), the index as a function of time t.

Level 3: $\overline{EMI(p)_{2.5}}$, averaged EMI for a period of time (p), i.e. a week or a month.

This is illustrated in Figure 3 of the manuscript. Therefore, P0 and p1 are two time periods defined for the comparisons of averaged meteorological impacts by EMI, i.e. p0 represents the month of January in 2015, and p1 represents the month of January in 2019. The p0 and p1 do not represent one period of time from p0 to p1. The focus of EMI applications is on the comparison of averaged meteorological difference between these two time intervals (p0 and p1).

For the first case of absolute stagnation, if $\Delta EMI = iTran + iAccu + iEmid = 1/c_0 \cdot Emis \cdot h$, as a constant, the equation (5) becomes for the period of p0 (n steps):

$$\overline{EMI(p0)_{2.5}} = EMI(p0) + (n - 1) \times \Delta EMI(p0) \times \Delta t$$

which means that averaged EMI will increase and be accumulated as the time goes on for n steps. This is exactly what this kind of meteorological conditions will bring about to the pollution levels.

In order to compare the difference of meteorological impacts, we have to define a new period, i.e. p1. If p 1 has the same initial conditions [EMI(p0)=EMI(p1)] and absolute stagnation, the averaged $\overline{EMI(p)_{2.5}}$ would be determined by the duration of the stagnation (m steps).

$$\overline{EMI(p1)_{2.5}} = EMI(p1) + (m - 1) \times \Delta EMI(p1) \times \Delta t$$

The longer of the stagnation, the larger of averaged $\overline{EMI(p)_{2.5}}$. This is exactly what $\overline{EMI(p)_{2.5}}$ is intended to be: a quantitative description of the meteorological impact on pollution levels. If n=m, the ratio of $\overline{EMI(p0)_{2.5}}/\overline{EMI(p1)_{2.5}} = 1$, indicating the same meteorological impact as expected.

If the period p1 is defined as the referee suggested: extreme dispersion conditions (assumed constant) with the same emission, we would expect a huge NEGATIVE $iTran$ for the $\Delta$EMI(p1), resulting $\Delta EMI(p1) \ll \Delta EMI(p0)$ and $\overline{EMI(p1)_{2.5}} \ll \overline{EMI(p0)_{2.5}}$ and reflecting favorite meteorological conditions for P1, i.e. the ratio of $\overline{EMI(p0)_{2.5}}/\overline{EMI(p1)_{2.5}} \gg 1$. Eventually, if

$\Delta EMI(\text{p1}) = iTran + iEmid < 0$, the dispersion would clean the pollutants for a certain period of time, and bring the EMI(p1) to reach zero as the concentration has reached zero by extreme dispersion; if $\Delta EMI = iTran + iEmid > 0$, we can then expect an increase of EMI(p1). The following figure illustrates the concept of the EMI and the areas below each curve (red and black line) is the averaged $\overline{EMI(p)}_{2.5}$ for each period, respectively, for the cases suggested by the referee (Figure 1).

[Figure]

Figure 1: Illustration of relationship between 3 levels of EMI definition for the cases suggested by Referee #2.

Please NOTE that since we used a constant emission at each location to compute the EMI, any changes in $\overline{EMI(p)}_{2.5}$ for two periods (p0 and p1) are solely attributed to the changes in meteorological conditions. If emission changes (+ or -) due to anthropogenic activities from p0 to p1 in conjunction with the meteorological variations, the $\overline{EMI(p)}_{2.5}$ will not be able to fully account for all the contributions, as the observational values are caused by both changes.

In order to separate the impacts of meteorology and emission contributions to the changes in pollutant concentrations from p0 to p1, the equations (6) and (7) are introduced and used to

quantitively assess the emission CHANGES only on the observed levels of pollutants (PM$_{2.5}$ for current study) from p0 to p1. An assumption is made here that under the same emissions at p1 (e1), the ratio of the averaged PM$_{2.5}$ concentrations under meteorology for p0 (m0) to the averaged PM$_{2.5}$ concentrations under meteorology for p1 (m1) is equal to the ratio of averaged EMI for each periods, i.e. $\overline{EMI(p0)_{2.5}} / \overline{EMI(p1)_{2.5}}$, which is exactly what EMI is intended to be. Therefor:

$$PM(m0, e1) \Big/ PM(m1, e1) = \overline{EMI(p0)_{2.5}} \Big/ \overline{EMI(p1)_{2.5}} \tag{6}$$

The impact of only emission changes from *e0* to *e1* on the concentration changes can be expressed as:

$$\Delta EMIS = \frac{PM(m0,e1) - PM(m0,e0)}{PM(m0,e0)} \times 100\% \tag{7}$$

where PM(*m0, e0*) and PM(*m1, e1*) are the observed concentrations at p0 and p1, respectively. PM(*m0, e1*) is estimated from Equation (6).

In summary, we think the assessment framework is solid based. The questions raised the referee was due to the confusion by the description part of methodology section, which may have misled the referee to derive and come out with the questions. Because of this, we have revised this section extensively to give a clearer description. Thanks for the referee.

Below are some technical corrections and comments:

(1) What are the units of EMI and $\Delta EMI$? Is EMI unitless as shown in Fig. 5/9? You will get different answers after doing dimensional analysis for Eqs. (1)-(3).

Reply: The EMI is unitless. Thanks to the referee who found the problems in Eqs (1)-(3): There is a term (1/h) missing in the equation and we have fixed them. The new equations are as follows:

$$iTran = \frac{1}{hC_0} \int_0^h \left( u \frac{\partial C}{\partial x} + v \frac{\partial C}{\partial y} + w \frac{\partial C}{\partial z} \right) dz$$

$$iAccu = \frac{1}{hC_0} \int_0^h \left[ \frac{\partial C}{\partial x} \left( Kx \frac{\partial C}{\partial x} \right) + \frac{\partial C}{\partial y} \left( Ky \frac{\partial C}{\partial y} \right) + \frac{\partial C}{\partial z} \left( Kz \frac{\partial C}{\partial z} \right) \right] dz$$

$$iEmid = \frac{1}{hC_0} \int_0^h \left[ Emis - (V_d + L_d) \right] dz$$

The calculations in the model was done with the right equations and therefore the results presented in the paper were not impacted by this problem.

(2) How to determine the initial value for EMI(t0)? Here I assume t0 denotes the first day of 2013, which is the start point of the model simulation. But the initial value for EMI is not mentioned in the manuscript.

Reply: In order to compare each year (or month) under the same conditions, the initial value of EMI(t0) was set the same for the first day of each year (or month). We also checked the sensitivity of EMI on the initial values of EMI(t0) and concluded that monthly averaged EMI was hardly impacted by the initial values. Nevertheless, the initial values for each month was set up by the averaged $PM_{2.5}$ concentrations for the first day from 2013 to 2019 divided by a constant C (35 um/m3). This has been added in the manuscript.

(3) What is $\hbar$ in Eq. (3)? Is it boundary layer height or not?

Reply: It is an arbitrary value of 1500 meters but it was a height defined to contain most of aerosol mass in the boundary layer.

(4) What kind of data were used for the correlation in Fig. 4? Monthly? Or Daily?

Reply: They are daily values used for the correlation.

(5) The time intervals for model evaluation are inconsistent throughout the manuscript. For example, Figs. 7-10 show the comparison from 2013 to 2019, but Table 2 shows the attribution results between 2015 and 2019.

Reply: There are two issues here that prompted us to use two different time intervals for the comparisons in the paper. The first issue is the completeness of the network observational data series of $PM_{2.5}$ in China. The systematical and network observations of $PM_{2.5}$ started in China from 2013. However, it took about two years (until 2015) to develop to the current status. Number of monitoring stations national-wide in 2013 was less than 900, reached to about 1400 in early 2015 and maintained the same up to now. To show the completeness of the observed PM2.5 time series and for most part of the paper, we made the comparison starting from 2013 as graph illustrations. The second issue is the data consistence and policy relevance of the assessment. Statistically, because the national observation site numbers are relative constant from 2015 to 2019, it makes more sense to use the 2015-2019 data for numerical assessment such as those shown in Table 2. The use of 2015-2019 data for Table 2 was also motivated by the introduction of the Environmental Protection Law of People's Republic of China in January 2015. For the regulation assessment point of view, the comparison Table 2 was in line with the date of the law introduction and the impact assessment by emission changes was more relevant to the interests of management to show how effective the law was.

---

## Referee Report (RR1)

Referee Report for "Assessment of meteorology vs control measures in China fine particular matter trend from 2013-2019 by EMI" by Sunling Gong et al. (manuscript #: acp-2020-348R)

In the response to my previous review, the authors added more information to try to support the validity of their EMI assessment framework. I appreciate their efforts and clarifications, but these additional information only confirms my previous concern about the fundamental flaw in this EMI framework. To illustrate my point more clearly, here I use two figures below to show why it's incorrect to add the emission term ($Emis$) in the $\Delta EMI$ and EMI calculation.

Again, let's revisit the first hypothetical case with extreme air stagnation conditions. In this case, we assume no advection ($iTran=0$), no turbulent diffusion ($iAccu=0$), no precipitation and wet deposition ($L_d = 0 \ in \ iEmid$), and negligible dry deposition compared with a constant emission ($Emis \gg V_d \ in \ iEmid$). Therefore, $\Delta EMI$ becomes

$$\Delta EMI = iTran + iAccu + iEmid = \frac{1}{hC_0} \int_0^h (Emis)dz = \frac{1}{C_0} Emis,$$

according to their updated Eq. (3) in the revised manuscript.

Based on this derivation and Eq. (1), EMI(t) would monotonically increase at a constant rate/slope of $\frac{1}{C_0} Emis$ from an initial value $EMI(t_0)$ along with time $t$ (Fig. 1).

$$EMI(t) = EMI(t_0) + \int_{t_0}^t \Delta EMI \times dt = EMI(t_0) + \Delta EMI \times (t - t_0).$$

Thanks to the clarification in the response, now $\overline{EMI(p_0)_{2.5}}$ becomes

$$\overline{EMI(p_0)_{2.5}} = \frac{1}{t_1-t_0} \int_{t_0}^{t_1} EMI(t)dt = \frac{1}{2}[EMI(t_0) + EMI(t_1)] = \frac{1}{2}[EMI(t_0) + EMI(t_0) + \Delta EMI \times (t_1 - t_0)] = EMI(t_0) + \frac{n}{2} \times \frac{1}{C_0} Emis \times \Delta t,$$

and $\overline{EMI(p_1)_{2.5}}$ becomes

$$\overline{EMI(p_1)_{2.5}} = \frac{1}{t_3-t_2} \int_{t_2}^{t_3} EMI(t)dt = \frac{1}{2}[EMI(t_2) + EMI(t_3)] = EMI(t_2) + \frac{m}{2} \times \frac{1}{C_0} Emis \times \Delta t,$$

given $p_0 = t_1 - t_0 = n \times \Delta t$ and $p_1 = t_3 - t_2 = m \times \Delta t$.

Note that the equations for $\overline{EMI(p_0)_{2.5}}$ and $\overline{EMI(p_1)_{2.5}}$ in the response are wrong due to incorrect time scaling factors ("$\frac{n}{2} \times \Delta t$" and "$\frac{m}{2} \times \Delta t$" here vs "$(n-1) \times \Delta t$" and "$(m-1) \times \Delta t$" in the response).

If assuming the same time interval length for $p_0$ and $p_1$ ($n = m$) as the authors did in the response, then

$$\frac{\overline{EMI(p_0)_{2.5}}}{\overline{EMI(p_1)_{2.5}}} = \frac{EMI(t_0)+\frac{n}{2}\times\frac{1}{c_0}Emis\times\Delta t}{EMI(t_2)+\frac{n}{2}\times\frac{1}{c_0}Emis\times\Delta t},$$ which can't be equal to 1 as they claimed in the response unless $EMI(t_0) = EMI(t_2)$.

However, the only way that satisfies $EMI(t_0) = EMI(t_2)$ is $t_2 = t_0$ because EMI(t) is a monotonically increasing function in this case based on the equations from the manuscript.

[Figure]

$$EMI(t) = EMI(t_0) + \int_{t_0}^{t} \Delta EMI \times dt,$$
$$where\ \Delta EMI = const > 0$$

Figure 1 the illustration of the EMI(t) function based on Eqs. (1)-(3) in the manuscript.

The above contradiction derives from the incorrect inclusion of Emis in iEmid/EMI as pointed out in the previous review. Let's see what will happen in the same stagnation case if excluding Emis in iEmid/EMI.

In this way, $iEmid = \frac{1}{hC_0}\int_0^h -(V_d + L_d)dz = 0$, and $\Delta EMI = iTran + iAccu + iEmid = 0$. Therefore, $EMI(t) = EMI(t_0)$ becomes a horizontal line with a constant value $EMI(t_0)$ for all time $t$ (Fig.2). Furthermore, $\frac{\overline{EMI(p_0)_{2.5}}}{\overline{EMI(p_1)_{2.5}}} = \frac{EMI(t_0)}{EMI(t_0)} = 1$ is satisfied for any time period $p_0$ and $p_1$. By applying this equation to Eqs. (6)-(7) in the manuscript, we can obtain

$$PM(m_0, e_1) = \frac{\overline{EMI(p_0)_{2.5}}}{\overline{EMI(p_1)_{2.5}}} \times PM(m_1, e_1) = PM(m_1, e_1),$$

and $\Delta EMIS = \frac{PM(m_0,e_1)-PM(m_0,e_0)}{PM(m_0,e_0)} \times 100\% = \frac{PM(m_1,e_1)-PM(m_0,e_0)}{PM(m_0,e_0)} \times 100\%$.

These two equations imply that the observed PM concentration change between $p_0$ and $p_1$ $(PM(m_1, e_1) - PM(m_0, e_0))$ is purely (100%) from the emission contribution in $\Delta EMIS$, while the contribution from meteorology $(1 - \Delta EMIS)$ is zero. This result is as expected considering all the assumptions in this case.

[Figure]

*Figure 2 the illustration of the proposed* EMI(t) *function without the inclusion of Emis in iEmid/EMI.*

The same problem also exists in the second extreme dispersion case, for which the illustration in Fig. 1 is still incorrect in the response. In this case, the EMI(t) function can't be a monotonically decreasing line with a constant slope as shown by Fig. 1 in the response because of the time variant concentration gradient in *iTran*. There are some other problems such as the arbitrary reset for the initial value of EMI($t_0$) in each year/month (see the answer to the second technical comment in the response). The low sensitivity of EMI to the initial value may result from highly variable meteorological conditions in the real atmosphere.

I'll stop here without further review, but I think the above examples are self-evident that the current EMI framework is flawed in its basis (mostly in *iEmid*). I suggest the authors think about their framework carefully and reconsider the submission of the manuscript in the current form.

---

## Author Response (AR2)

Thanks for the referee's comments on this manuscript. After carefully reading the comments, we have made following point-to-point replies:

In the response to my previous review, the authors added more information to try to support the validity of their EMI assessment framework. I appreciate their efforts and clarifications, but these additional information only confirms my previous concern about the fundamental flaw in this EMI framework. To illustrate my point more clearly, here I use two figures below to show why it's incorrect to add the emission term (*Emis*) in the ΔEMI and EMI calculation.

**Reply:** First of all, the EMI was defined to reflect the impact of meteorology on the PM pollution in different regions and should be comparable with the in-situ observations. The addition of emissions term in the EMI was made to show the impact difference of meteorology on PM at different regions. We also would like to use an extreme case to explain the necessity to include emission term in EMI in order to compare with observations at different locations. Still under the hypothetical case with extreme air stagnation conditions as suggested by the referee, we choose two locations: one with an emission of Emis as a constant and another without emission, i.e. Emis=0. In reality, the location with a constant emission would be subject to a heavy pollution episode while the location without emissions would be pollution free. If we used the suggestion by the referee without emission, the EMIs at both locations would be equal, which could not reflect the difference of the real situations at these two locations and could not be compared with real observations. By introducing the emissions in the EMI, the difference would be clearly shown: the location with emissions would be experiencing an accumulation of pollution with time as indicated by ΔEMI=1/C0 Emis; the location without emission would be pollution free with time as indicated by ΔEMI=0, which mimics the real situations as EMI was intended to be.

As we understood from pollution formation mechanisms, the pollutant emission was the fundamental cause of any pollution and the meteorology was the external force to modulate the pollution strength. The EMI was defined to show the impact of meteorology on PM pollution under a constant emission for any locations by including the emissions. If there were no emissions, there were no pollution at all and no meaning to define an EMI.

As a matter of fact, there exists no incorrectness or correctness in defining an EMI, all depending on the purposes of the EMI applications and the targets to compare.

Again, let's revisit the first hypothetical case with extreme air stagnation conditions. In this case, we assume no advection ($iTran=0$), no turbulent diffusion ($iAccu=0$), no precipitation and wet deposition ($L_d = 0$ in $iEmid$), and negligible dry deposition compared with a constant emission ($Emis \gg V_d$ in $iEmid$). Therefore, $\Delta EMI$ becomes

$$\Delta EMI = iTran + iAccu + iEmid = \frac{1}{hC_0}\int_0^h (Emis)dz = \frac{1}{C_0}Emis,$$

according to their updated Eq. (3) in the revised manuscript.

Based on this derivation and Eq. (1), EMI(t) would monotonically increase at a constant rate/slope of $\frac{1}{C_0}Emis$ from an initial value $EMI(t_0)$ along with time $t$ (Fig. 1).

$$EMI(t) = EMI(t_0) + \int_{t_0}^t \Delta EMI \times dt = EMI(t_0) + \Delta EMI \times (t - t_0).$$

Thanks to the clarification in the response, now $\overline{EMI(p_0)}_{2.5}$ becomes

$$\overline{EMI(p_0)}_{2.5} = \frac{1}{t_1-t_0}\int_{t_0}^{t_1} EMI(t)dt = \frac{1}{2}[EMI(t_0) + EMI(t_1)] = \frac{1}{2}[EMI(t_0) + EMI(t_0) +$$
$$\Delta EMI \times (t_1 - t_0)] = EMI(t_0) + \frac{n}{2} \times \frac{1}{C_0}Emis \times \Delta t,$$

and $\overline{EMI(p_1)}_{2.5}$ becomes

$$\overline{EMI(p_1)}_{2.5} = \frac{1}{t_3-t_2}\int_{t_2}^{t_3} EMI(t)dt = \frac{1}{2}[EMI(t_2) + EMI(t_3)] = EMI(t_2) + \frac{m}{2} \times \frac{1}{C_0}Emis \times \Delta t,$$

given $p_0 = t_1 - t_0 = n \times \Delta t$ and $p_1 = t_3 - t_2 = m \times \Delta t$.

Note that the equations for $\overline{EMI(p_0)}_{2.5}$ and $\overline{EMI(p_1)}_{2.5}$ in the response are wrong due to incorrect time scaling factors ("$\frac{n}{2} \times \Delta t$" and "$\frac{m}{2} \times \Delta t$" here vs "$(n-1) \times \Delta t$" and "$(m-1) \times \Delta t$" in the response).

**Reply:** Thanks to the referee. The referee was right here that we did use an incorrect time scaling factor as we missed one initial step in Equation 5 using the time step from 0 to n-1 to represent n time steps. This has been corrected in the revised manuscript.

If assuming the same time interval length for $p_0$ and $p_1$ ($n = m$) as the authors did in the response, then

$$\frac{\overline{EMI(p_0)}_{2.5}}{\overline{EMI(p_1)}_{2.5}} = \frac{EMI(t_0)+\frac{n}{2}\times\frac{1}{C_0}Emis\times\Delta t}{EMI(t_2)+\frac{n}{2}\times\frac{1}{C_0}Emis\times\Delta t},$$ which can't be equal to 1 as they claimed in the response unless $EMI(t_0) = EMI(t_2)$.

However, the only way that satisfies $EMI(t_0) = EMI(t_2)$ is $t_2 = t_0$ because EMI(t) is a monotonically increasing function in this case based on the equations from the manuscript.

**Reply:** This is exactly what EMI is intended to be used: using the same initial conditions to compare the EMI for two segments of time periods at the same location. One case study in the manuscript was the comparison of January of 2013 ($p_0$) and January of 2016 ($p_1$), where the same initial conditions and emissions were used to quantify the meteorological impact. Please note that in the model simulations, we have set the $EMI_{t_0}$ and $EMI_{t_2}$ equal to each other and differences in simulated averaged $EMI(p_0)$ and $EMI(p_1)$ would indicate the impacts of meteorology on pollutants (Fig. 1). If the same meteorology occurred in p0 and p1, the $EMI(p_0)$ and $EMI(p_1)$ would be the same as expected.

[Figure]

Figure 1: EMI simulation schemes for two periods of time.

The above contradiction derives from the incorrect inclusion of Emis in iEmid/EMI as pointed out in the previous review. Let's see what will happen in the same stagnation case if excluding Emis in iEmid/EMI.

In this way, $iEmid = \frac{1}{hc_0} \int_0^h -(V_d + L_d)dz = 0$, and $\Delta EMI = iTran + iAccu + iEmid = 0$.

Therefore, $EMI(t) = EMI(t_0)$ becomes a horizontal line with a constant value $EMI(t_0)$ for all time $t$ (Fig.2). Furthermore, $\frac{EMI(p_0)_{2.5}}{EMI(p_1)_{2.5}} = \frac{EMI(t_0)}{EMI(t_0)} = 1$ is satisfied for any time period $p_0$ and $p_1$.

By applying this equation to Eqs. (6)-(7) in the manuscript, we can obtain

$PM(m_0, e_1) = \frac{EMI(p_0)_{2.5}}{EMI(p_1)_{2.5}} \times PM(m_1, e_1) = PM(m_1, e_1)$,

and $\Delta EMIS = \frac{PM(m_0,e_1)-PM(m_0,e_0)}{PM(m_0,e_0)} \times 100\% = \frac{PM(m_1,e_1)-PM(m_0,e_0)}{PM(m_0,e_0)} \times 100\%$.

These two equations imply that the observed PM concentration change between $p_0$ and $p_1$ ($PM(m_1, e_1) - PM(m_0, e_0)$) is purely (100%) from the emission contribution in $\Delta EMIS$, while the contribution from meteorology ($1 - \Delta EMIS$) is zero. This result is as expected considering all the assumptions in this case.

**Reply:** Again, if the case was the extreme air stagnation conditions, our formulation would arrive at the same conclusions by including the same emission for the two periods, i.e., $EMI(p_0)_{2.5}$ $=EMI(p_1)_{2.5}$, as same initial conditions were used for n=m. However, we would not see any PM concentration change between p0 and p1 as the emissions, initial conditions and meteorology were all the same. This is what we should expect for the two periods. If no emissions were included, we would not see any increase of EMI with time and the acclamation of any pollutant in any period

would not be reflected in EMI, which contradicted to the real PM situations and made the EMI not comparable with PM observations.

The same problem also exists in the second extreme dispersion case, for which the illustration in Fig. 1 is still incorrect in the response. In this case, the EMI(t) function can't be a monotonically decreasing line with a constant slope as shown by Fig. 1 in the response because of the time variant concentration gradient in *iTran*. There are some other problems such as the arbitrary reset for the initial value of EMI($t_0$) in each year/month (see the answer to the second technical comment in the response). The low sensitivity of EMI to the initial value may result from highly variable meteorological conditions in the real atmosphere.

I'll stop here without further review, but I think the above examples are self-evident that the current EMI framework is flawed in its basis (mostly in *iEmid*). I suggest the authors think about their framework carefully and reconsider the submission of the manuscript in the current form.

**Reply:** If the same degree of extreme dispersion occurred for both p0 and p1, we should also arrive at EMI(p0) =EMI(p1) but with decreasing trends for both periods as the same initial conditions and emissions were used for both p0 and p1. The Fig. 1 in the last version of response showed a monotonically decreasing line simply due to the assumption of a constant extreme dispersion for the sake of explaining the framework. In reality, the decreasing line is definitely variable as controlled by the time variant concentration gradients as indicated by the referee.

Finally, it is evident that the inclusion of emission term in the EMI was not a flaw but a necessity for the right application of EMI. However, we would thank the effort of the referee whose comments and suggestions have made this manuscript much more scientifically accountable.

---

## Author Response (AR3)

The manuscript developed an environmental meteorology index (EMI) and use this index to further quantify the contributions of meteorology and emission control to the air quality improvement in China from 2013 to 2019. A major concern raised by a previous reviewer is about the inclusion of emission in the definition of EMI. After reviewing the manuscript, review comments and the authors' response, I sided with the previous reviewer, and think it is a valid concern regarding the emission term in the calculation of EMI. This issue needs to be thoroughly addressed.

1. Including Emis (emission) in their eq. (3) is important for diagnosing accumulation potentials or dispersing potentials, as illustrated in Fig. 3 and discussed in the last paragraph of page 14 and the first paragraph of page 15. Without emis, it would be hard to see how pollutant accumulates or disperse, as the accumulation or dispersion depends not only on meteorology but also on emissions. So for the purpose of calculating accumulation or dispersing potentials, it is fine to include Emis in Eq. (3). In this regard, I agreed with the authors that the exact definition of EMI will depend on scientific objectives the index is used to address.

A: Thank you for agreeing with this point.

2. But for the current manuscript, the most important goal is to diagnose contributions of meteorology and emission control to the air quality improvement in China. In the manuscript, Eq. (6) is used to calculate the sole contribution from emission control. One critical term in Eq. (6) is PM (m0, e1), the PM concentration under conditions of unchanged meteorology at p0 but with new emission e1. To get this, the authors then used the ratio of EMI(p0)_bar to EMI(p1)_bar (Eq. (7)). And the authors claim that this ratio can be used to "reflect the impact ratio of sole meteorology variations on the concentrations between p0 and p1 with the same emission at p1" (page 16, lines 10-12). But this argument is flawed, as the formula of EMIS includes emission contribution (Emis in Eq. (1)), as pointed out by the previous reviewer. As emission differences also contribute to the difference in EMI between p0 and p1, the authors can not attribute the ratio EMI(p0)_bar/EMI(p1)_bar to the pure impact of meteorology. So the primary objective of this manuscript, I agreed the previous reviewer and do not think it is appropriate to include the emission term in EMI.

A: We agree partially with the reviewer's comments. Let's clarify each term in Eq. 7 to better explain the assumption used in the manuscript. PM(m1, e1) is the OBSERVED PM concentrations at p1 with emission e1 and meteorology m1. PM(m0, e1) is a hypothetically non-measurable quantity, indicating the PM concentration at p1 with emission e1 and meteorology m0, that does not exist in reality but needs to assess the sole impact of emission changes. EMI(p0) and EMI(p1) are the simulated quantity of equivalent PM concentrations at p0 and p1, respectively, with a fixed emission (Emis in Eq. 3) but with separate meteorology of m0 and m1. If we re-write the Eq. 7 as follows:

$$\frac{EMI(p0)_{2.5}}{EMI(p1)_{2.5}} = \frac{PM(m0, e1)}{PM(m1, e1)}$$

which assumes that under the same emissions, the ratio of EMIs under two meteorology (m1, m0) equals to the ratio of PM concentrations under the same two meteorology (m1, m0). This is another reason that the emission term (Emis) is needed in the EMIs, otherwise the above equation (assumption) cannot be derived. We agreed with the reviewer's comment that "emission differences also contribute to the difference in EMI between p0 and p1". It is true that, in reality, the changes in PM concentrations or EMIs are not a linear addition of changes in emission and meteorology as assumed in Eqs. 6 and 7. This limitation has been included in Section 2.4 and Conclusions.

3. It is also important for the authors to quantitatively evaluate the effectiveness of the proposed EMI index for the attribution. For example, it would significantly strengthen this manuscript if the authors can use their CTM to calculate the contribution of emission control to the air quality improvement by perturbing emissions, and then use the CTM results to evaluate the results diagnosed from EMI.

A: Yes. We have done this evaluation and have added the results in the manuscript. Following texts and a Table was added to the manuscript.

The applicability of EMI to assess the meteorology and emission changes is also evaluated by results from a full chemical transport model (MM5/CUACE) and observational data for $PM_{2.5}$ in China for Novembers of 2017 and 2018. The averaged $EMI_{2.5}$ and observational data for the two months were used to estimate the emission change ratio (E-Ratio in Table 2) by Equations 6-7 from 2017 to 2018. In order to evaluate the correctness of this emission change estimate, the E-Ratio was used to adjust the emissions for November 2018 from the base emissions of the same month for 2017, which were then implemented in the MM5/CUACE to simulate the $PM_{2.5}$ concentrations for the two months, respectively. If the simulated concentration differences (M-Ratio) for the two months were comparable with the observed concentration differences (O-Ratio), it can be concluded that the emission change estimated by the EMI framework was reliable and could approximately represent the actual emission changes. Table 2 summarizes the analysis results of this evaluation for six typical cities. It is clear that the O-Ratios for the six cities are very comparable with M-Ratios, indicating that the EMI framework can be reasonably used to estimate the emission changes over time.

Table 2: Comparison of PM$_{2.5}$ Concentrations in Novembers of 2017 and 2018 from Ambient Observations and from CTM Simulations by EMI-derived Estimated Emission Changes

| City | EMI$_{2.5}$ | | Observations | | | Emission Changed | CTM Simulated | | |
|---|---|---|---|---|---|---|---|---|---|
| | 2017 | 2018 | 2017 | 2018 | O-Ratio | E-Ratio | 2017 | 2018 | M-Ratio |
| Beijing | 1.8 | 3.6 | 45.7 | 72.8 | 1.59 | 0.80 | 42.3 | 67.5 | 1.59 |
| Shanghai | 2.7 | 2.6 | 42.0 | 40.1 | 0.95 | 1.00 | 52.7 | 51.2 | 0.97 |
| Jinan | 3.3 | 4.9 | 57.1 | 85.8 | 1.50 | 1.02 | 62.4 | 90.9 | 1.46 |
| Xian | 2.4 | 2.7 | 94.8 | 84.7 | 0.89 | 0.79 | 95.1 | 86.9 | 0.91 |
| Zhengzhou | 4.3 | 6.2 | 73.9 | 100.4 | 1.36 | 0.96 | 80.4 | 91.1 | 1.13 |
| Shenyang | 1.8 | 2.7 | 40.2 | 48.9 | 1.21 | 0.82 | 73.3 | 120.1 | 1.63 |